# A maize mutant in the glutamate receptor-like *dwarf13* is modified by *cis*-acting natural variation and a *cornichon* homolog

Amanpreet Kaur[1,2], Rajdeep S. Khangura[3], Brian P. Dilkes[1,2]*

1 Department of Biochemistry, Purdue University, West Lafayette, Indiana, United States of America,
2 Center for Plant Biology, Purdue University, West Lafayette, Indiana, United States of America,
3 Department of Plant and Agroecosystem Sciences, University of Wisconsin-Madison, Madison, Wisconsin, United States of America

* bdilkes@purdue.edu

## Abstract

Deciphering the molecular basis of complex traits requires understanding how natural genetic variation interacts with underlying biological pathways. In this study, we explored how natural genetic variation influences traits in maize affected by a semi-dominant maize dwarfing allele, *Dwarf13–1 (D13-1)* which encodes a defective ionotropic glutamate receptor (GLR). This allowed us to investigate natural genetic variation in the genome affecting GLR signaling in maize. We implemented an F1 association mapping (FOAM) approach, where heterozygous mutants carrying the semi-dominant *D13-1* allele were crossed with a maize association panel. The resulting F1 families segregated 1:1 for mutant and wild-type phenotypes allowing comparisons between the congenic F1 hybrid siblings to identify and map natural alleles that interact with the *D13-1* mutant allele. FOAM mapping detected two loci that modify the expression of the *D13-1*/+ mutant phenotype. The phenotypic impacts of both loci were epistatically controlled by *D13-1*, and only affected the phenotypes of mutant F1 hybrids. One, *tropotriskaideka1* (*tod1*), encoded a maize homolog of the GLR-interacting *cornichon* gene and modified *D13-1*/+ mutant severity. A second, encoded by the *d13* locus itself, affected the severity of the *D13-1*/+ phenotype via variation in the wild-type allele in the heterozygous mutants. By integrating gene expression analyses, these epistatic interactions, and SNP linkage information we identified multiple, unlinked, alleles affecting expression of the wild-type D13 transcript that modify mutant trait expression. Greater expression of the wild-type D13 allele increased plant height and suppressed *D13-1*/+ mutant severity, consistent with a multi-subunit complex GLR structure and complex-poisoning mode-of-action for the semi-dominant *D13-1* allele. This approach identifies natural alleles affecting the GLR pathway in maize and establishes GLRs and their interactors as dose-dependent regulators of plant architecture. Our pathway-focused framework and epistasis testing of natural variants provides greater confidence in identifying genes contributing to complex traits.

**Data availability statement:** All relevant data are within the manuscript and its Supporting information files.

**Funding:** This work was supported by United States Department of Energy Office of Science (BER) Grants DE-SC0023305 and DE-SC0020368 and National Science Foundation grant IOS-2309932 to B.P.D and USDA NIFA Postdoctoral Fellowship Award # 2022-67012-36601 to R.S.K.. The funders had no role in study design, data collection and analysis, decision to publish, or preparation of the manuscript.

**Competing interests:** The authors have declared that no competing interests exist.

## Author summary

Complex traits are shaped by interactions among multiple genes. In this study, we explored how natural genetic variation influences traits in maize affected by a semi-dominant maize dwarfing allele, *Dwarf13–1/+*, which encodes a defect in a glutamate receptor-like gene. We crossed *D13-1/+* with diverse maize lines creating an F1 association mapping population. This enabled us to use matched wild-type and mutant siblings as case-control pairs to identify natural variants that enhance or suppress mutant phenotypes. Unlike most QTL and GWAS approaches, F1 association mapping uses the identity of the mutant gene to discover natural variants that are epistatic modifiers of mutant alleles and assign these variants to a molecular pathway. Through this approach, we identified maize homologs of the Drosophila *Cornichon* gene and cis-regulatory variation at *d13* itself as major modifiers of the mutant phenotype.

## Introduction

One of the great achievements of molecular genetics is the identification of genes with no prior link to growth and development due to the presence of a mutant allele affecting a phenotype of interest. Mutants provide links between genotype and phenotype by enabling the examination of the direct effects of variation in a single gene on phenotypes. Plant dwarf mutants have been instrumental in identifying the molecular nature of diverse mechanisms that regulate plant growth and development, beginning with the founding observations of genetics in the dwarf pea mutants studied by Mendel [1–4]. In maize, mutations in genes underlying phytohormone biosynthesis, transport, and signaling pathways have resulted in numerous dwarf mutants that have illustrated the effects of these hormones on plant growth and development [5–12]. Studies of the interactions between dwarf mutants affected in the perception and biosynthesis of hormones have revealed extensive crosstalk between these hormones, that influence both plant growth and development [8,12–16].

We previously demonstrated that the maize dwarf mutant *D13-1* was a novel semi-dominant allele encoded by a point mutation in a glutamate receptor-like ion channel [17]. *D13-1* is the first dwarf mutant identified among the GLR receptors, however, its role in regulating height remains unclear, and no hormone signaling pathway has been implicated [17]. Plant glutamate receptor-like (GLR) channels are homologs of ionotropic glutamate receptors (iGluRs) that are non-selective ligand-gated ion channels found across the tree of life and best known for mediating neurotransmission in mammals [18–21]. GLRs assemble as tetramers to form a functional ion channel [22]. In plants, GLR $Ca^{2+}$ channels participate in light signaling, carbon-nitrogen metabolism, defense responses, stomatal closure, pollen tube growth, root growth and development, and drought response [18–20,23–38]. Plant GLR structures have been determined [22] and structural similarities to iGluRs include domains of known biophysical function including a gating domain the inner aperture

of the channel that is known to regulate $Ca^{2+}$ influx [39,40]. Multiple characterized mutants in the conserved gating domain disrupt the hinge activity, resulting in a constitutively open and hyperactive channel, as demonstrated by electrophysiological studies, such as in the *lurcher* mutant of a mouse GLR [41–43]. The *D13-1* mutant allele results from an alanine to threonine change at a residue conserved in all glutamate receptors [17] that is within the channel gating domain [44].

One goal for the discovery of gene functions is to develop understanding of the mechanisms affecting trait variation in living systems. However, the molecular identification of causal genes for natural variation and the assignment of natural variants to genetic and biochemical pathways remain major challenges in biology. Knowing the biological pathway that a natural variant modulates to affect variation in a trait, can accelerate the discovery of the gene responsible for a natural allele. One way to do this is by studying interactions between natural alleles and mutants of known function. The interactions between mutant genes and natural variation in populations simultaneously discovers alleles in populations and annotates them as interacting with a known pathway or process [45–48]. This helps gain information about the pathways and mechanisms by which natural variation affects phenotypic variation in species. By using a mutant with a molecularly identified causative locus, correct specification of the gene encoding the interacting natural variant becomes more efficient, facilitating discovery and characterization a molecular pathway's role in affecting natural variation in a trait [47–50]. Dominant and semi-dominant alleles are particularly powerful tools for this approach because a heterozygote can be crossed to diverse germplasm to generate an F1 association mapping (FOAM) population which can be phenotyped directly. Each F1 family is comprised of isogenic *D13-1/+* and wild-type siblings that differ only for the mutant locus. This permits case-control comparisons within each F1 hybrid family to identify alleles that enhance or suppress the mutant phenotypes while controlling for the effect of genetic background via the wild-type sibling phenotypes. When the molecular identity of the mutant gene is known, these epistatic interactions can be annotated as acting in the pathway(s) affected by the mutant. Integrating these data with the effects of genetic variation on gene expression can provide further insights into molecular mechanisms underlying the altered phenotype [51,52] and aid in the identification of causative genes [53–60] from natural variation.

In this study, we crossed the *D13-1/+* mutant to an association panel representing the global diversity of maize. This identified multiple loci encoding natural alleles affecting mutant phenotype expression. FOAM mapping identified a regulator of *D13-1/+* phenotypic severity at *tropotriskaideka1* (*tod1*), a natural variant linked to a maize gene encoding a homolog of the GLR-interacting CORNICHON family of proteins. We also demonstrate the existence of strong *cis*-acting regulatory variation affecting D13 transcript abundance, which also influenced *D13-1* phenotypic severity. *Cis*-regulatory variation that increased *d13* expression reduced mutant phenotypic severity, suggesting that higher concentrations of wild-type subunits restore normal function to the multimeric channel encoded by this locus. This work links both GLRs and the interacting cornichon family of proteins to plant height and derives molecular mechanistic insight from the combination of multiple data types in GWAS experiments.

## Results

### Phenotypic variation in *D13-1/+* FOAM population

An F1 association mapping (FOAM) population was generated by crossing heterozygotes carrying the semi-dominant *D13-1* allele in the B73 genetic background (*D13-1/+*:B73) with a panel of 224 diverse maize lines (Fig 1A). The *D13-1* and wild-type alleles segregated 1:1 within each isogenic F1 family, permitting the observation of background effects on the severity of the *D13-1/+* phenotype by directly comparing F1 hybrid siblings that differed at the *d13* locus. The F1 families were planted in the field in two replications and phenotyped for plant height and a subjective visual mutant severity score. Plant height was measured as flag leaf height (FlHT) and ear height (EaHT) at maturity in both mutant and wild-type siblings from 224 isogenic F1 hybrid families (Fig 1B). Prior work demonstrated suppression of dwarfism and other gross morphological defects of *D13-1/+* by alleles contributed by the Mo17 inbred in B73 x Mo17 F1 hybrids [17]. Consistent with these prior observations, we observed several F1 families which completely suppressed the mutant phenotype,

PLOS Genetics

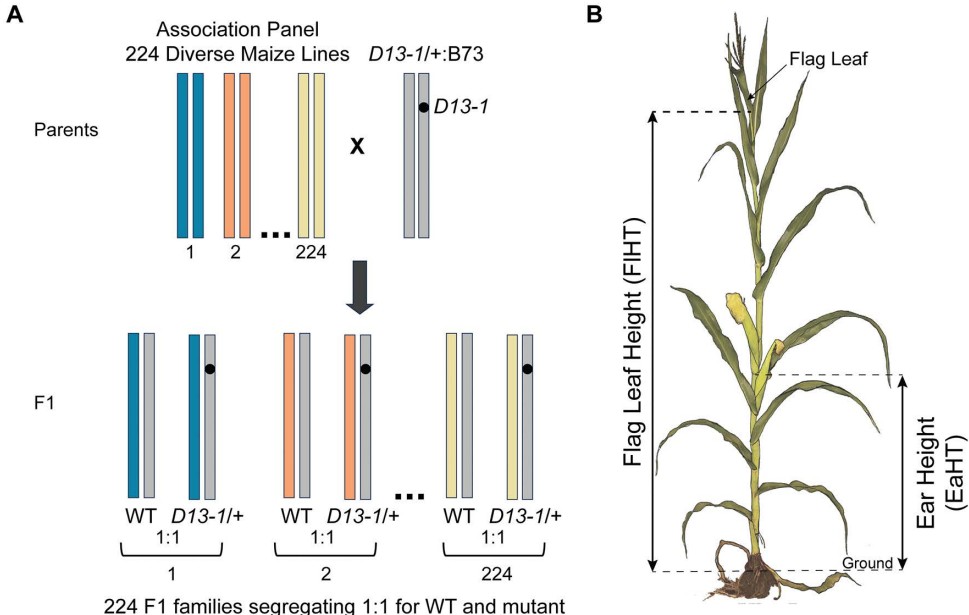

**Fig 1. Design of experiment. A)** Schematic of crossing scheme used to develop *D13-1* F1 association mapping population (FOAM). Each pair of vertical bars represents chromosome 5. The chromosomes from diverse maize inbred lines are represented in different colors. *D13-1* allele is depicted by a black circle. The F1 progeny from each cross segregates 1:1 for mutant (*D13-1*) and isogenic wild-type (WT) siblings. Each pair of homologous chromosomes in the F1 progeny comprises one inbred chromosome and one B73 chromosome with or without the *D13-1* allele; **B)** Height measurements were recorded as flag leaf height (FlHT) and ear height (EaHT) for mutant and wild-type siblings at maturity in each F1 family.

and *D13-1/+* mutants were indistinguishable from the wild-type siblings (Fig 2A, S1 Table). To estimate the trait values for mutants in the suppressing F1 families, we employed the conservative approach of measuring the most wild-type individuals and three randomly selected individuals assigned as mutants for height (FlHT and EaHT). Trait means were compared between the two replicates to confirm suppression. In addition to the strong genetic suppression, phenotypic instability of the *D13-1/+* mutants was reported within the B73 background [17]. As an approach to minimize the effect of trait instability on mapping, we used the height of the most extreme dwarf individual in each F1 family to determine the shortest mutant flag leaf height (SFHT). Measurements were taken in two replications, and average SFHT was calculated for each F1 family. This approach should minimize errant variance in trait means if there is trait instability within the F1 families, as was previously observed in the inbred B73 background [17]. In addition, the severity of phenotypic effects of *D13-1* in F1 families was assessed by assigning a severity score ranging from 0-5 to the mutant plants in each F1 family at six weeks after planting. Mutants showing normal, wild-type-like plant height were assigned a score of 0, whereas those with the most severe height reduction and tillering were assigned a score of 5 (Fig 2A, 2B).

The FlHT of wild-type plants among all the F1 families was normally distributed (Anderson-Darling test p-value >0.05) and ranged from 170 cm to 312 cm with an average of 242.7 ± 24.6 cm (Fig 2C, Tables 1, S1). The FlHT of *D13-1/+* mutants was variable across the F1 families and ranged from 11.7 cm to 285.7 cm. The EaHT of WT plants in F1 families ranged from 63.3 cm to 186 cm, and the EaHT of *D13-1/+* mutants varied from 2.5 cm to 167.5 cm (Fig 2D, Tables 1, S1). EaHT and FlHT were strongly correlated within the wild-type ($r^2 = 0.90$) and mutant F1 hybrids ($r^2 = 0.93$) (Fig 3). SFHT among the F1 families showed a similar distribution to mutant FlHT (MT FlHT), ranging from 9.5 cm to 284 cm (Fig 2E). The average mutant heights (MT FlHT) and average shortest mutant heights (SFHT) for the F1 families were highly correlated ($r^2 = 0.99$, Fig 3). The high correlation between replicate plantings indicated that phenotypic instability within these families is not a large contributor to variance in trait expression.

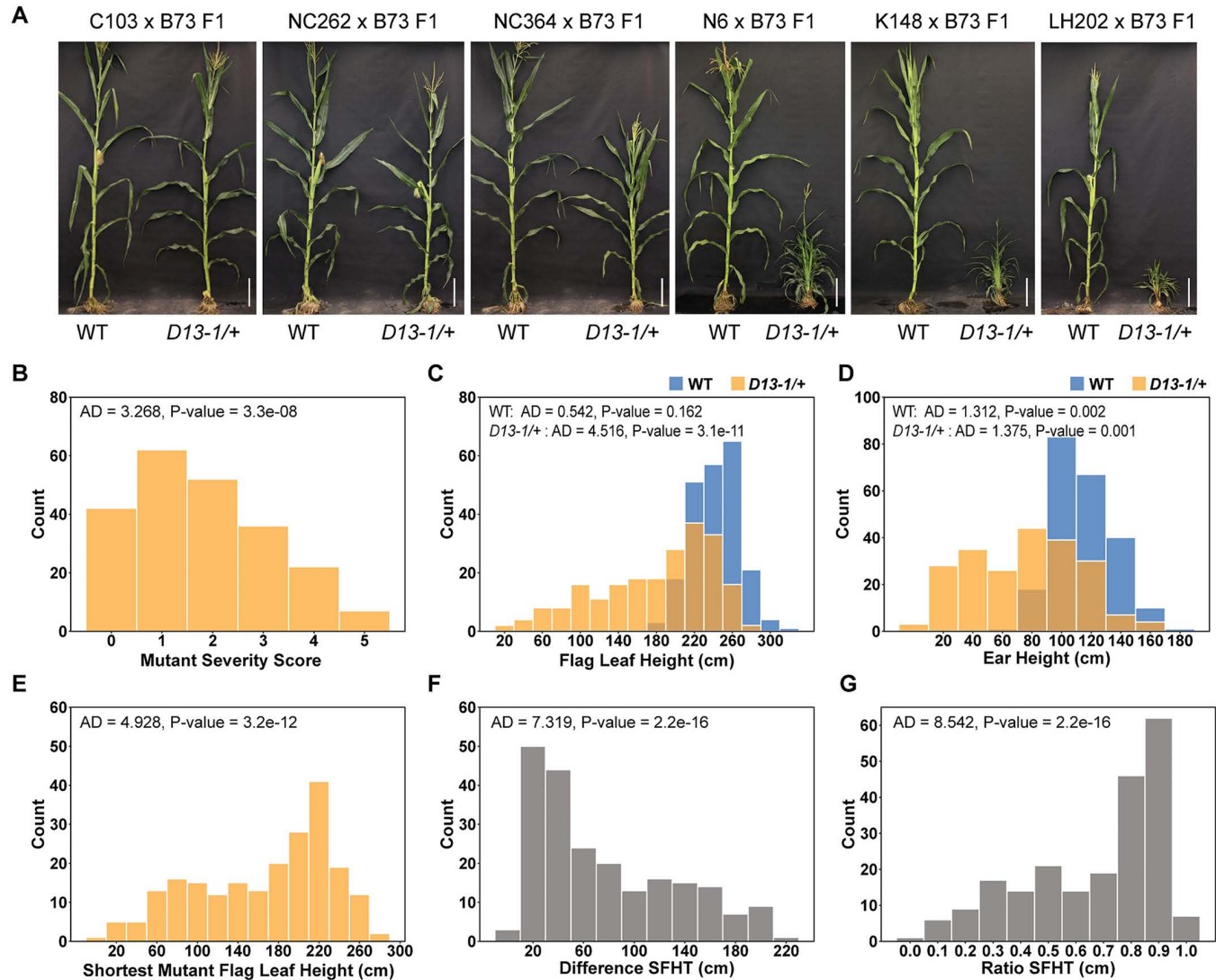

**Fig 2. Phenotypic variation in *D13-1* FOAM population. A)** The height of *D13-1/+* mutants (right) as compared to their isogenic wild-type (WT) siblings (left) at maturity in different F1 families, scale bar = 29 cm; **B)** Frequency distribution of mutant severity score among the F1 families; **C)** Frequency distribution of mutant and WT flag leaf heights (FlHT) in FOAM population. Blue bars represent the distribution of WT heights, and gold bars represent mutant heights; **D)** Frequency distribution of WT and mutant ear heights in the F1 families; Frequency distributions of **E)** Shortest mutant flag Leaf Height (SFHT), **F)** Difference SFHT, and **G)** Ratio SFHT among the F1 families. The height measurements are reported in centimeters (cm). Each distribution was tested for normality using the Anderson-Darling (AD) test. P-value <0.05 indicates that data are not normally distributed.

The heights of wild-type and mutant siblings in the F1 families showed a strong positive correlation (FlHT $r^2 = 0.64$, SFHT $r^2 = 0.61$, EaHT $r^2 = 0.60$; Fig 3). In prior experiments, the Mo17 background completely suppressed the phenotype of *D13-1/+* and *D13-1/D13-1* genotypes [17]. We observed strong suppression again in the F1 families, with *D13-1/+*:Mo17/B73 receiving an average score of 1.5 and exhibiting ~20 cm difference between the tallest and shortest plants in that family. In the FOAM population, the *D13-1/+* phenotype was completely suppressed in 48 lines, including Mo17, with a score ≤ 1.5 and difference FlHT ≤ 25.4 cm. Removing the F1 families resulting from *D13-1/+* crosses to these 48 lines decreased the correlation between mutant and wild type heights only slightly, to $r^2 = 0.55$, but they were still

**Table 1. The phenotype of *D13-1/+* mutants in FOAM population.**

| Trait | Genotype | Trait Mean | Standard Deviation | Range | Heritability (H²) |
|---|---|---|---|---|---|
| Flag Leaf Height (FlHT) | WT | 242.73 | 24.56 | 170.3-312 | 0.801 |
| | *D13-1/+* | 179.14 | 60.90 | 11.7-285.7 | 0.903 |
| Ear Height (EaHT) | WT | 114.62 | 20.0 | 63.3-186 | 0.889 |
| | *D13-1/+* | 75.01 | 36.87 | 2.5-167.5 | 0.875 |
| Mutant severity score | *D13-1/+* | 2.03 | 1.30 | 0-5 | 0.765 |
| Shortest Mutant FlHT (SFHT) | *D13-1/+* | 165.96 | 66.86 | 9.5-284 | 0.894 |

Height trait measurements reported here are in centimeters (cm).

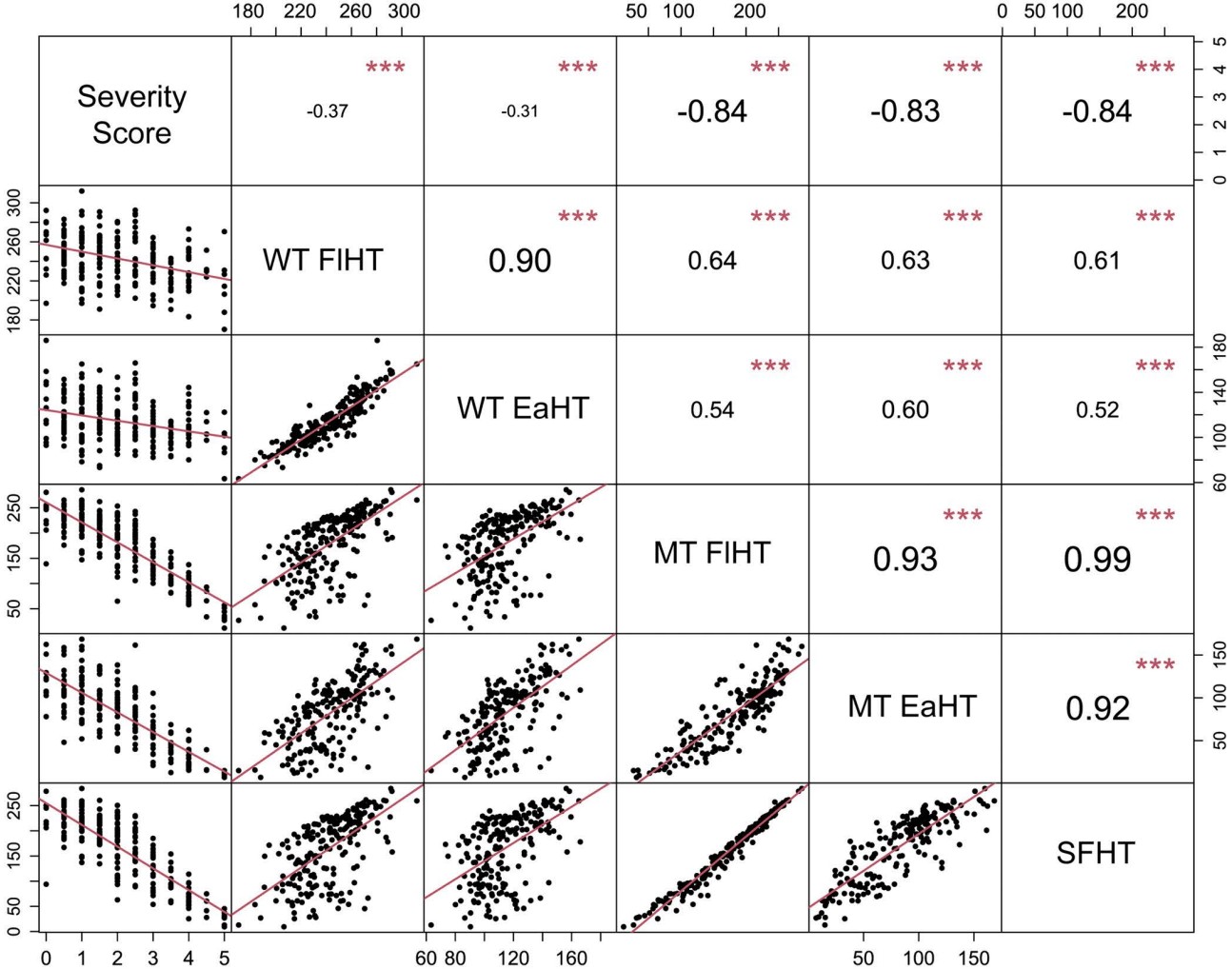

**Fig 3. Correlation of *D13-1* phenotypes in F1 families.** Pearson's correlation matrix depicting six traits from the *D13-1* FOAM population including mutant severity score, flag leaf height (FlHT) and ear height (EaHT) in mutant (MT) and wild-type (WT) siblings, and shortest mutant flag leaf height (SFHT). Values above the diagonal represent pairwise correlation coefficients among the traits. Asterisks denote significance with ***p-value <0.001. Scatter plots below the diagonal display bivariate relationships, with red lines representing best-fit linear regression.

significantly correlated indicating that genetic factors altering plant height in the wild type contributed substantially to variation in the mutant plant height. To control for variation in height affected by genetic background, and more accurately estimate the effect of genetic background on the *D13-1/+* mutant phenotype, we treated each family as a set of case-control observations and calculated the difference between WT and mutant heights and ratio of mutant to WT heights for each F1 family (Fig 2F, 2G). As expected, the mutant phenotypic severity score exhibited a strong negative correlation with mutant height traits ($r^2$ from -0.83 to -0.84). Remarkably, mutant severity scores also showed a weak negative correlation with wild type height ($r^2$ ranging from -0.31 to -0.37; Fig 3). The strong positive correlation between mutant and wild type height and the negative correlation with score suggests that genetic mechanisms affecting plant height variation in wild-type maize hybrids also altered the phenotypic severity of *D13-1/+*. This could be due to the additive effects of these alleles on both mutant and wild-type hybrids, though some non-linearity is visible in the relationship between mutant and wild type heights. The broad-sense heritability estimates for all traits were greater than 0.7 (Table 1), indicating a strong genetic control of these traits.

### Genome-wide Association study (GWAS) identifies regulators of *D13-1* phenotype

A GWAS was performed to identify natural alleles that affected variation in the mutant severity score and plant height traits from 224 F1 families using 22.4M SNPs. At a genome-wide Bonferroni corrected p-value threshold of $2.2 \times 10^{-9}$, a single locus was detected affecting the mutant severity score (Fig 4A, 4B). The SNP 6–70758267 was the top SNP associated with the *D13-1* severity score at a p-value of $7.65 \times 10^{-10}$. We named this locus *tropotriskaideca1* (*tod1*), meaning modifier of 13 (Greek-derived roots *trópos* "to turn, change" and *triskaideka* "thirteen"). The *tod1* locus also exhibited associations with mutant height traits (SFHT, MT FIHT, MT EaHT), and their ratio and difference traits; however, the associations with quantitative height assessments did not pass the Bonferroni threshold (Figs 4C, 4D, 4E, S1, S2 Table). We examined the *tod1* locus and observed strong LD from 70607277 bp to 71539073 bp (Fig 4F). The *d13* gene encodes a glutamate receptor-like ion channel (GLR) [17]. We examined this window for other GLRs and homologs of the only known GLR-interacting proteins in plants, the CORNICHON HOMOLOG (CNIH) proteins [61]. The CORNICHON protein was first identified in Drosophila [62] and its mammalian and plant homologs interact with glutamate receptors to facilitate the sorting and activation of these channels [61,63]. Remarkably, a gene encoding a cornichon homolog protein (*tod1*, v3: GRMZM2G073023, v4: Zm00001d036084, v5: Zm00001eb270040) was present within this region.

To further test if the gene Zm00001d036084 is the causative gene at the *tod1* locus, we investigated if the phenotypic variation affected by natural variation at *tod1* could be attributed to variation in the expression of the Zm00001d036084 gene. We analyzed the effects of 526 SNPs identified within the *tod1* locus (S2 Table) on TOD1 (Zm00001d036084) transcript abundance. Transcript abundance data for the Zm00001d036084 gene in the inbreds constituting the maize association panel were obtained from four above-ground tissues: germinating shoot (GShoot), the base of the third leaf (L3Base), and mature leaves collected during the day (LMAD) and night (LMAN)] described in a previous study [64]. Of the 526 SNPs associated with *D13-1* phenotype, 355 affected Zm00001d036084 transcript accumulation in at least one of the four tissues at p-value ≤ $1 \times 10^{-4}$ (S3 Table). The top SNP, 6-70758267, for variation in mutant severity score was also associated with Zm00001d036084 transcript abundance in GShoot, L3Base, and LMAN (S3 Table). The alleles associated with increased Zm00001d036084 transcript accumulation in GShoot, L3Base, and LMAD increased the severity of the mutant phenotype (S3 Table). The same alleles that affected Zm00001d036084 transcript accumulation in these three tissues also affected accumulation of these transcripts in LMAN but resulted in decreased accumulation of transcript. These SNP effects and the switch in direction demonstrated the interaction of Zm00001d036084 *cis*-regulatory polymorphisms with diurnal *trans*-acting regulation. The expression of Zm00001d036084 was greatest during the day (Mean Box-Cox transformed counts in GShoot: 4.6; L3Base: 3.6; LMAD: 10.4) while the gene was barely expressed during the night (Mean Box-Cox transformed counts in LMAN: 0.6). These results demonstrate a relationship between *cis* variation in Zm00001d036084 expression and the mutant phenotype

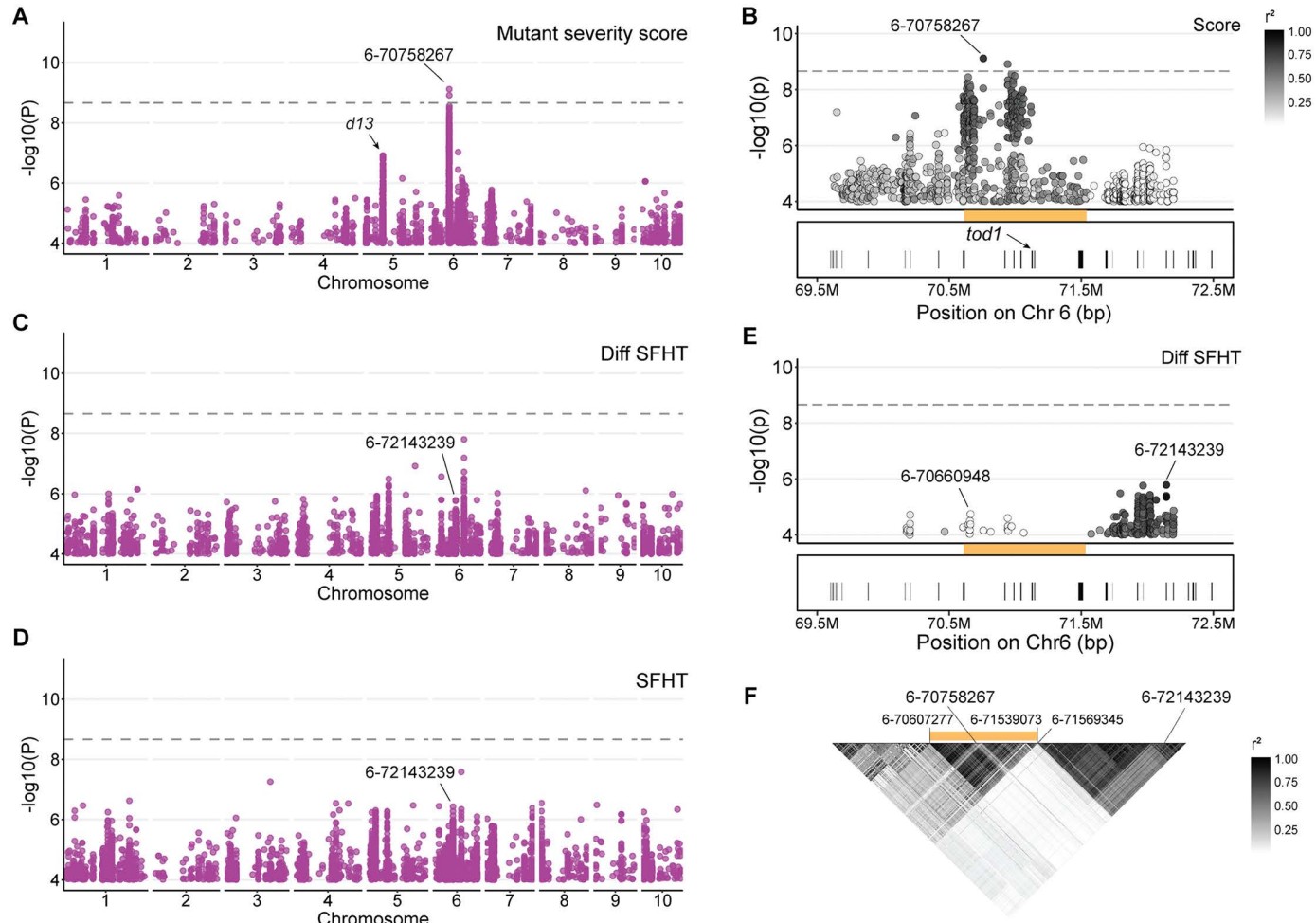

**Fig 4. Natural variation at *tod1* modifies *D13-1* phenotype. A)** SNPs associated with *D13-1* mutant severity score at p-value ≤ 1x10⁻⁴. Grey dashed line represents the genome-wide Bonferroni corrected p-value threshold. The top SNP associated with the mutant severity score is labelled; **B)** SNPs at *tod1* locus associated with mutant severity score. The gold bar represents the location of the *tod1* locus. Within this region, a *cornichon* homolog was designated as *tod1* gene. The SNPs are colored based on the degree of linkage disequilibrium (r²) with the top SNP (6-70758267); **C)** SNPs associated with Diff SFHT at p-value ≤ 1x10⁻⁴. The SNP 6-72143239 is the top SNP in this region affecting Diff SFHT but is unlinked to *tod1*; **D)** SNPs associated with SFHT at p-value ≤ 1x10⁻⁴); **E)** Locus plot for SNPs at *tod1* associated with Diff SFHT. The SNPs are colored based on r² with 6-72143239. The SNP 6-70660948 is the top SNP affecting Diff SFHT within the *tod1* locus; **F)** LD plot for Hapmap3 SNPs on chromosome 6 associated with *D13-1* phenotypic traits. The SNPs within the *tod1* locus are tightly linked.

severity, providing compelling evidence that Zm00001d036084 plays a functional role in driving the phenotype associated with these genetic variants and encodes the *tod1* gene. This is consistent with prior studies in mammals and plants that showed CNIH proteins interact with GLRs [61,63].

Maize has a second homolog of CNIH protein, also located on chromosome 6, that was identified from the PLAZA Monocots 5.0 database [65]. We examined the associations at this other CNIH (*cnih1*, v3: GRMZM2G018885, v4: Zm00001d035235, v5: Zm00001eb264160) for significant effects on the *D13-1/+* phenotype as a candidate gene association. There were 5056 Hapmap3 SNPs within ± 250kb of *cnih1* in our dataset, and a Bonferroni corrected p-value threshold for the candidate gene association was 9.9x10⁻⁶. None of the SNPs within the *cnih1* locus passed this statistical threshold for any trait (S4 Table), suggesting that natural variation at *cnih1* does not regulate *D13-1* phenotype.

None of the SNP-trait associations for the mutant height traits or the ratio and difference traits exceeded the genome-wide Bonferroni threshold or a commonly used arbitrary threshold of $10^{-8}$ (S5 Table). However, we evaluated candidate genes for all SNPs associated with mutant severity score and mutant height traits, including ratio and difference traits at p-value ≤ $1x10^{-4}$ (S6 Table). The current lack of information about the regulation and ligands of the GLRs in plants makes it difficult to infer which candidate genes encode the causative alleles for these associations.

## Natural variation at the *d13* locus affects the *D13-1* phenotype

The *D13-1* allele is semi-dominant, suggesting that the wild-type *d13* allele from the association panel parent might affect the phenotypic severity of F1 *D13-1/+* heterozygotes. Although such a modifier was not detected in our *D13-1* FOAM study at the genome-wide Bonferroni threshold, we looked for the possibility of a similar mechanism using a single locus test. To test this, we assessed if the SNP variation at *d13* locus was linked to the variation in mutant severity score and plant height. We tested the 6248 SNPs within a ± 250 kb window of the *d13* gene for genetic associations with trait variation. With 6248 tests, all associations with p-value ≤ $8x10^{-6}$ would exceed a Bonferroni corrected α < 0.05 threshold. SNP-trait associations for 50 SNPs exceeded this threshold for severity score, mutant flag leaf height traits (SFHT, MT FIHT, MT EaHT), ratio traits (ratio SFHT, ratio FIHT, ratio EaHT), or difference traits (difference SFHT, difference FIHT) (Fig 5A, 5B, S7 Table). Among these 50 SNPs, 31 affected the score along with at least one of the above mutant height traits at a p-value ≤ Bonferroni threshold. The alleles associated with lower scores were consistently linked to increased mutant height (Fig 5C). At a slightly more permissive threshold of p-value ≤ $1x10^{-4}$, we identified 202 SNPs within a ± 250 kb window of the *d13* gene associated with variation in severity score, mutant heights, ratio, or difference traits (S7 Table). Of these 202 SNPs, 66 were associated with at least two traits (Fig 5, S7 Table) with 47 SNPs passing the Bonferroni threshold for at least one trait. The alleles associated with the scores and heights of the mutants always resulted in lower scores and greater heights moving together (Fig 5C). None of the SNPs that affected the severity score, and mutant height were associated with wild type height traits, indicating that variation in *d13* itself is unlikely to contribute to wild type height variation in maize. A single SNP at this locus, that was not associated with mutant traits even at the permissive threshold, was linked to ear height in wild-type plants with p-value = $3.18x10^{-5}$ but it does not exceed the Bonferroni threshold (S7 Table). These data demonstrate that there is consequential natural variation at *d13* that affects the severity of the *D13-1/+* phenotype.

## *Cis*-acting natural variation linked to *d13* expression affects the severity of the *D13-1* phenotype

To determine if the effects of natural variation at the *d13* locus on the *D13-1* phenotype were linked to the variation in expression of the *d13* gene, we conducted an eGWAS. Transcript abundance data for the *d13* gene in the maize association panel were obtained from four above-ground tissues: GShoot, L3Base, LMAD, and LMAN [64]. The data from these tissues capture D13 transcript abundance at different growth stages in maize. *Cis*-acting SNPs were defined as those within 50 kb of the transcription start and stop sites for the *d13* gene (S8 Table). Strong *cis*-acting alleles at the *d13* locus were detected in all four tissues (Fig 6A, 6B, 6C, 6D, S8 Table), indicating that *d13* expression is regulated by *cis*-acting natural variation. There were a total of 1632 SNPs located within ± 50 kb window around the *d13* gene in our dataset. Of these, 171 SNPs passed a Bonferroni corrected p-value threshold of $3.1x10^{-5}$ (1632 tests; α = 0.05) for this locus. We determined the effects of these *cis*-acting natural variants affecting D13 transcript abundance on mutant phenotypes. Of these 171 SNPs, 45 affected *D13-1* severity score and/or height traits at a p-value ≤ $1x10^{-4}$, which exceeds the Bonferroni threshold for 171 tests (Fig 6E). At a relaxed threshold of p-value ≤ $1x10^{-4}$, 206 SNPs were associated with D13 transcript abundance at *d13* locus (S9 Table) but no additional associations with *D13-1* phenotypic traits were detected. The alleles associated with higher D13 transcripts were associated with lower score and/or taller mutants. None of these 206 SNPs affected wild type height traits, consistent with *D13-1* epistatically enhancing the natural variants. These data demonstrate that lower accumulation of D13 transcripts correlates with increased severity of the mutant phenotype.

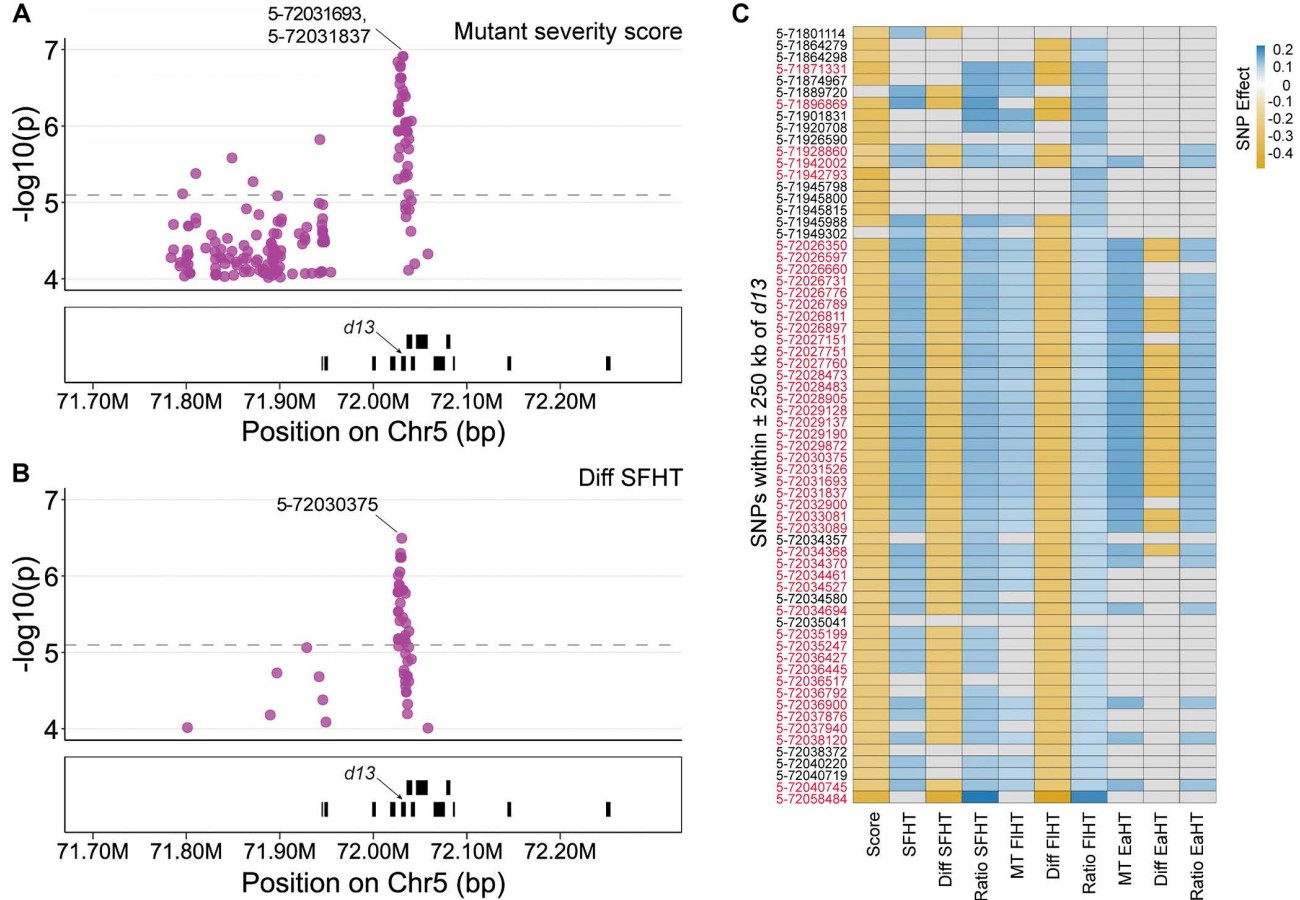

**Fig 5. Natural variation at *d13* influences *D13-1* phenotypes.** SNPs within ±250kb of *d13* gene associated with **A)** Mutant severity score and **B)** Difference of shortest mutant flag leaf height from wild type flag leaf height (Diff SFHT). The top SNPs associated with each trait are labelled. The grey dashed line is the Bonferroni corrected significance threshold based on SNPs in this locus; **C)** Effects of the SNPs at *d13* locus on *D13-1/+* phenotypic traits. Only the SNPs associated with at least two traits at p-value ≤ 1x10⁻⁴ are displayed here, and those marked in red passed the Bonferroni-corrected p-value threshold for at least one trait.

If the expression level of the wild-type D13 protein suppresses the mutant phenotype, then we should observe perfect correspondence between transcript level and phenotypic trait values across the alleles at this locus. Using the historical recombination in the maize association panel, we tested if the D13 transcript abundance and *D13-1* phenotype perfectly co-segregate or if recombination occurs between these traits. From the 1632 SNPs located within ±50 kb window around the *d13* gene, we identified the SNPs associated with D13 transcript abundance and/or phenotype at p-value ≤ 1x10⁻⁴. At this threshold, 66 SNPs affected the transcript levels of *d13* in GShoot and 50 affected *D13-1* score (Fig 7A, S9 Table). The analysis of the effects of 44 shared SNPs affecting *d13* expression and score showed that alleles associated with reduced accumulation of D13 transcripts always increased the mutant score (Fig 7B, S9 Table). Similarly, we identified 44 SNPs associated with difference SFHT (S9 Table). Of these, 41 also affected D13 transcript levels in GShoot. The effects of all SNPs that affected variation in both D13 transcript levels and SFHT were consistent with reduced expression in seedling shoots decreasing height in the mutants at maturity (S9 Table). A linkage disequilibrium (LD) analysis was performed for all the *cis*-acting SNPs at the *d13* locus that significantly affected the D13 transcript abundance in GShoot and/or the SNPs that affected mutant phenotype.

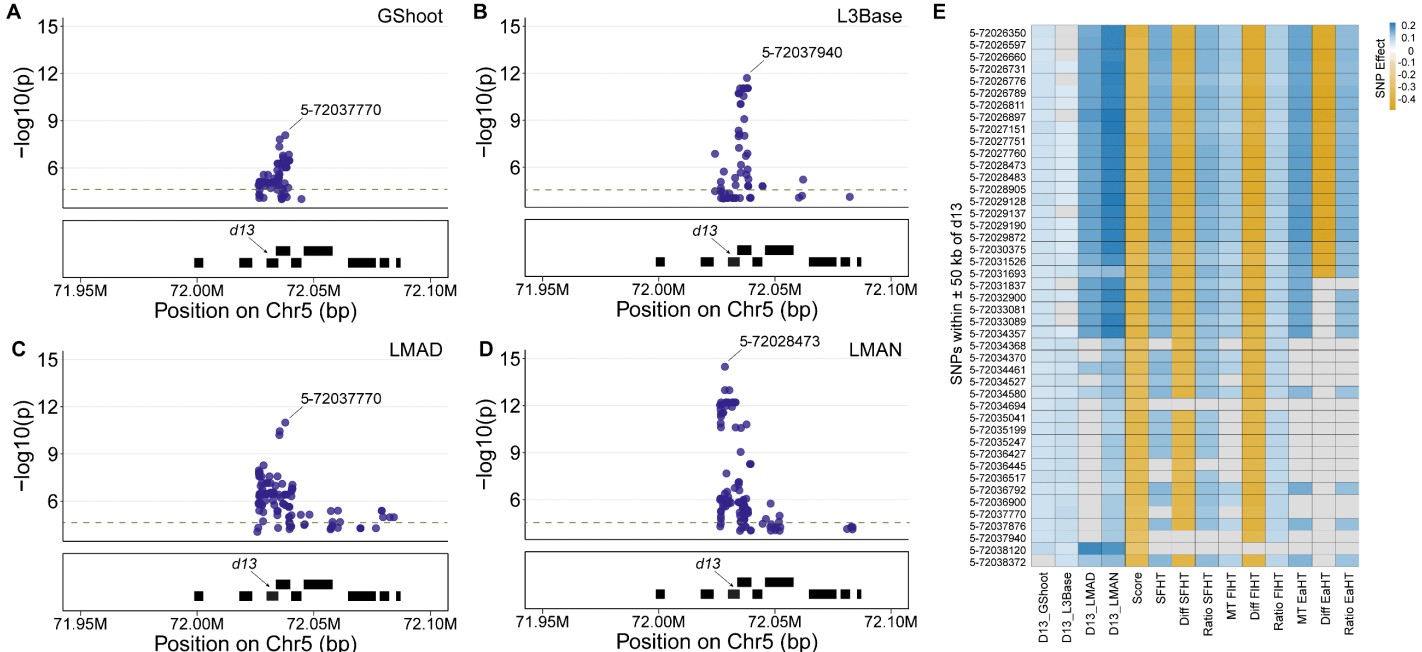

**Fig 6. Variation in D13 transcript abundance is associated with *D13-1*/+ phenotypic variation.** *Cis* SNPs within ±50kb of the *d13* gene affecting D13 transcript abundance in **A)** Germinating shoot (GShoot), **B)** Base of the third leaf (L3Base), **C)** Mature leaf collected during day (LMAD), and **D)** Mature leaf sampled at night (LMAN). Top SNPs associated with D13 transcript abundance in each of the four tissues are labelled. Grey dashed line represents the Bonferroni-corrected significance threshold (p-value = 3.1x10⁻⁵); **E)** Effects of the *cis* SNPs at *d13* locus on D13 transcript abundance and *D13-1*/+ phenotypic traits. SNPs that affected both the D13 transcript abundance in at least one tissue (p-value ≤ 3.1x10⁻⁵) and at least one of the *D13-1* phenotypic traits (p-value ≤ 1x10⁻⁴) are displayed here.

SNPs that affected both the traits (D13 transcript levels and mutant phenotype) were in high LD (Figs 7C, S2), as expected from the observation of no recombinants altering the effect directions on the two traits. *Cis* SNPs that only affected D13 transcript abundance were not in high LD with SNPs that affected both the traits, indicating the presence of multiple alleles and demonstrating that expression level in germinating shoot was not sufficient to determine *D13-1*/+ phenotype. SNPs that had a significant effect only on phenotype and affected expression in the opposite direction were unlinked to SNPs that affect both traits in the same direction. Clustering of the SNPs based on LD showed at least four *cis*-regulatory alleles in this locus affected D13 transcript accumulation (Figs 7D, S2). Similar results were observed for SNPs at the *d13* locus that were associated with D13 transcript abundance in L3Base, LMAD, and LMAN and *D13-1*/+ trait expression. For example, in the comparison of D13 abundance in L3Base and the mutant severity score, multiple alleles, with opposite effect directions, segregate as unlinked blocks, indicating at least two haplotypes with effects on both expression and trait, and additional independently segregating alleles affecting only transcript abundance (S3 Fig). Variants affecting only the abundance of D13 and not score or height, that were additionally unlinked from SNPs jointly affecting expression and *D13-1*/+ phenotypes, indicate that there were multiple alleles affecting gene expression and that *cis*-acting expression polymorphism was not sufficient to change the phenotype of *D13*/+ plants. The exact nature of the difference between these alleles is not yet known. These data demonstrate that *cis*-regulated variation in *d13* expression is not solely responsible for variation in plant height or severity score, and multiple *cis*-regulatory alleles at *d13* are segregating in *Z. mays*. These results are similar to our previous FOAM study using chlorophyll biosynthetic mutant where multiple *cis*-allele haplotypes at the locus encoding the mutant tester were identified [48].

PLOS Genetics

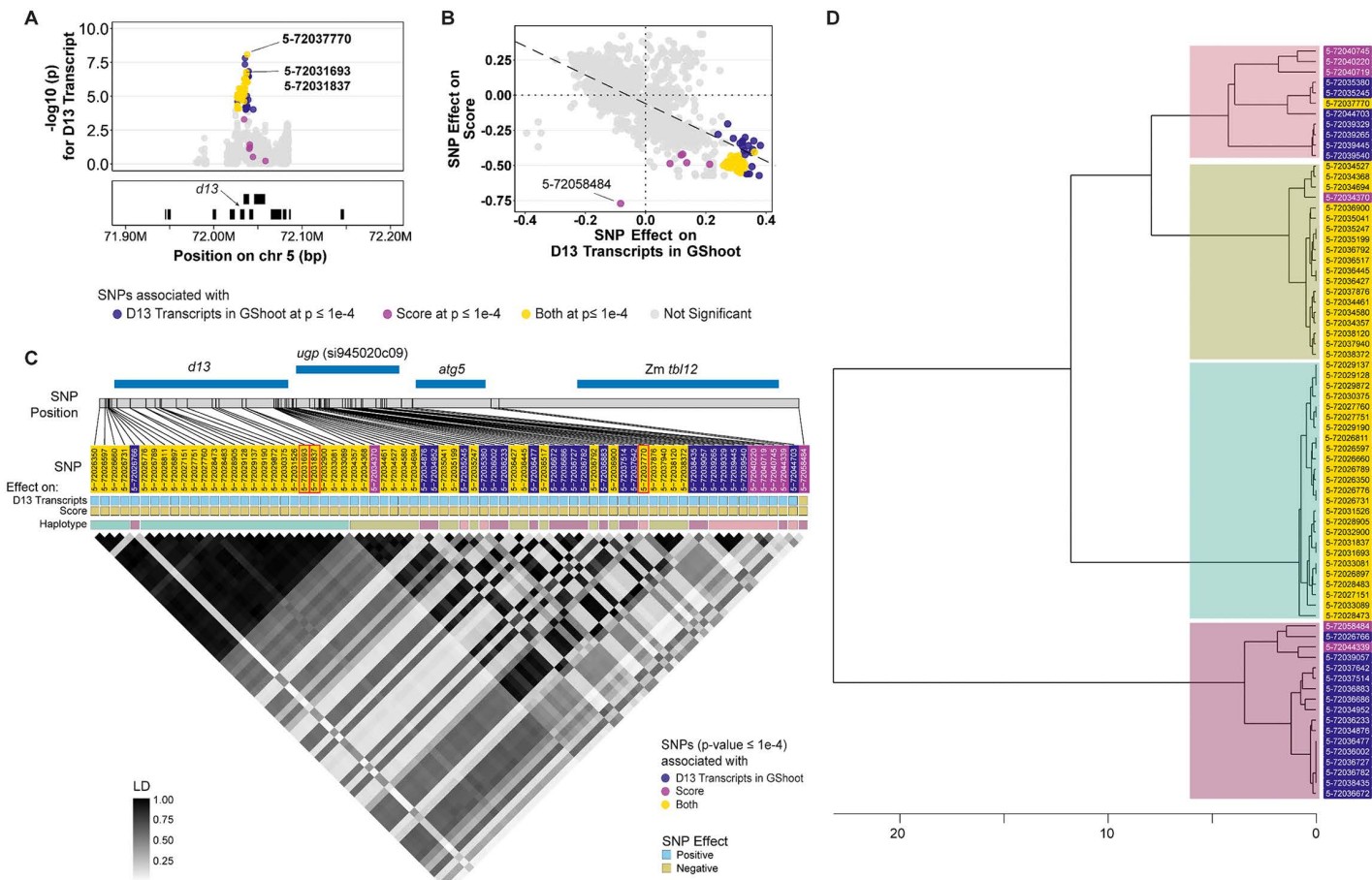

**Fig 7. Multiple *cis*-regulatory alleles at *d13* locus affect the D13 transcript abundance and mutant severity score. A)** SNPs within ±50 kb of *d13* gene associated with D13 transcript abundance in GShoot and/or Mutant severity score at p-value ≤ 1x10⁻⁴. Grey dots represent SNPs that were not significantly associated with either of the two traits at p-value ≤ 1x10⁻⁴. The top SNPs associated with both traits are labelled; **B)** Scatter plot depicting the relationship between SNP effects on D13 transcripts in GShoot and Mutant severity score. Dashed trend line indicates a negative correlation between SNP effects on two traits; **C)** LD Plot for SNPs within ±50 kb of *d13* gene affecting D13 transcript abundance in GShoot and/or Mutant severity score at p-value ≤ 1e-4. Blue bars at the top represent gene models. SNPs are highlighted based on trait associations at p-value ≤ 1e-4. Boxes below the SNPs depict the direction of effects on traits, where blue indicates a positive effect and yellow indicates a negative effect of SNP on the trait. Colored bars below these mark different haplotypes; **D)** Hierarchical clustering of SNPs based on LD estimates (r²). Colored boxes represent different haplotypes.

## Natural variation at other maize GLRs weakly modified the *D13-1* phenotype

Glutamate receptors are multimeric proteins that assemble as tetramers to form functional ion channels [22,39,44]. In maize, there are 18 GLRs, each potentially contributing to the formation of functional ion channels (S10 Table). However, the interactions between different GLRs to form the tetrameric channel remain largely unknown. A co-expression analysis of D13 transcript accumulation using publicly available co-expression data (database ATTED-II version 12.0) identified *glr7* as strongly co-expressed with *d13* (S4 Fig). The *d13* and *glr7* genes are syntelogs that arose due to the recent whole genome duplication in the ancestry of maize [17] and are likely to carry out identical biochemical functions. We mined the phenotypic GWAS results to determine if natural variation at any maize GLRs, besides *d13,* affected the *D13-1* phenotype. A total of 83175 SNPs within 250kb of all GLRs, except *d13,* were tabulated and tested for associations with *D13-1* phenotypic traits. None of these SNPs, including SNPs linked to *glr7*, were associated with any trait at an α < 0.05

Bonferroni adjusted significance threshold (p-value = 6x10$^{-7}$) for this SNP set (S11 Table). However, if the threshold was relaxed to p-value ≤ 1x10$^{-4}$, suggestive associations were identified at *glr7* (Zm00001d054016) for mutant height traits and epistatic effects (Diff SFHT, Ratio SFHT, SFHT, Diff FlHT, Ratio FlHT and Diff EaHT). In addition, suggestive SNP-trait associations for phenotypic traits in the FOAM population at p-value ≤ 1x10$^{-4}$ were identified at *glr1* (Zm00001d033769), *glr2* (Zm00001d002532), and four locations with tandemly arranged GLR genes; *glr3* (Zm00001d005782) and *glr4* (Zm00001d005783); *glr8* (Zm00001d014451), *glr9* (Zm00001d014456), and *glr10* (Zm00001d014458); *glr12* (Zm00001d018329) and *glr13* (Zm00001d018332); *glr16* (Zm00001d020563) and *glr17* (Zm00001d020568). SNPs at *glr2* and at the tandemly arrayed genes *glr8, glr9* and *glr10* were associated with both wild type and mutant height traits. SNPs at the tandemly arranged *glr3* and *glr4* were associated with WT FlHT and Diff FlHT. SNPs at three loci containing *glr1, glr12, glr13, glr16* and *glr17* only affected WT EaHT. Future work investigating the interactions between GLRs will be necessary to determine whether these proteins interact and whether these genetic associations are true positives.

We examined the SNPs within 250kb of all GLRs for their effects on D13 transcript accumulation to investigate if alleles of other GLRs encode *trans*-acting effects. This identified three SNPs linked to *glr14* (Zm00001d018614) and *glr15* (Zm00001d018615) on chromosome 7 that affected D13 transcript abundance in GShoot at a Bonferroni adjusted p-value ≤ 6x10$^{-7}$ (S12 Table). At a relaxed threshold of p-value ≤ 1x10$^{-4}$, additional associations were identified at five loci encoding six glutamate receptors, including *glr2, glr5* (Zm00001d006508), *glr6* (Zm00001d052064), *glr7* on chr 4, and two tandemly arranged GLRs *glr12* and *glr13* on chr 5 (S12 Table). A single SNP at *glr2* was associated with D13 transcript abundance in LMAN and a single SNP at *glr5* was associated with D13 transcript accumulation in L3Base. SNPs at *glr6* were associated with D13 transcript abundance in GShoot and mature leaf tissue LMAN. SNPs at locus containing *glr12* and *glr13* were associated with D13 transcript abundance in all four tissues: GShoot, L3Base, LMAD, and LMAN. The SNPs at *glr7* also affected D13 transcript abundance in all four tissues. Variation linked to *glr7* and *glr12/glr13* also had weak effects on the *D13-1/+* phenotype (see above; S11 Table). This raises the possibility that, in addition to encoding a syntelog and being co-expressed with D13, *glr7* affects the *D13-1/+* mutant phenotype by regulating the accumulation of D13 transcripts.

We performed an eGWAS for all maize GLRs using the maize association panel. Of the 17 GLRs, expression data were available for 12 GLRs in the eGWAS data set [64]. This identified *cis* variation affecting transcript accumulation for 11 GLRs (S13 Table). In addition to the strong *cis*-regulatory allele at *d13*, even stronger *cis*-regulatory effects were identified at *glr5* and the tandemly duplicated *glr8* and *glr9* (S13 Table). To determine if *cis*-acting variation affecting the expression of other maize GLRs altered *D13-1* traits, we tabulated the *cis*-acting SNPs and tested their effects on the *D13-1* phenotypic traits. Unlike D13, none of the *cis* SNPs affecting variation in any of the other GLRs had any effects on the phenotype of *D13-1* mutants, even at a permissive p-value ≤ 1x10$^{-4}$ threshold. This indicated that the *D13-1* phenotype was sensitive to variation in the expression of *d13* but not to variation affecting the expression of these other GLRs. This suggests that plant genomes may encode a diversity of GLR complexes, with different paralogs involved in discrete and independent signaling events.

### Admixture GWAS

Modern maize is derived from the admixture of two teosinte subspecies *Zea mays* spp. *mexicana* and *Zea mays* spp. *parviglumis*. We used the admixture estimates for all alleles in the maize association panel to determine if the variation in *D13-1* phenotype in F1 families was associated with alleles contributed to *Zea mays* spp. *mays* from one of these two ancestors. We used *Z. mays* spp *mexicana* and *Z. mays* spp *parviglumis* as training populations to estimate admixture in 224 diverse maize lines and performed a GWAS using the Bayesian estimate of admixture at each SNP as genotype in the association test. The ancestry estimates are not independent between SNPs but are derived from joint estimations from multiple linked positions. As such the appropriate multiple test correction is unclear. At the *d13* locus, no ancestry-estimates were associated with any *D13-1* phenotype even at a permissive p-value cutoff of 1x10$^{-4}$. This indicates that the alleles at *d13* that affect trait variation are unlikely to be the result of a differential contribution from either ancestral population.

## Discussion

### The *D13-1* mutant demonstrates that GLRs are regulators of plant architecture

Plant GLRs are pivotal in $Ca^{2+}$-mediated signal transduction, influencing a wide array of biological processes, including carbon-nitrogen metabolism, stomatal aperture movement, pollen tube growth, and root development [24–28,37,38]. GLRs also play a significant role in responses to biotic and abiotic stresses [29–32,34,66–69]. In addition to these roles, GLRs are emerging as regulators of plant architecture. The EMS-induced single base pair change in the maize *D13-1* mutants results in dwarf phenotypes characterized by shorter internodes and reduced meristem size as compared to the wild-type siblings [17]. These phenotypes are also observed in F1 families obtained by crossing the *D13*-1/+ mutant to diverse maize lines, highlighting the role of *d13* in growth regulation. Similarly, knockout alleles of a rice GLR, OsGLR3.4, exhibit dwarf phenotypes with erect leaves and reduced internode length [35], mirroring brassinosteroid-deficient phenotypes. Notably, OsGLR3.4 was identified as a direct target of OsBZR1, a key transcription factor in brassinosteroid (BR) signaling, and plays a role in BR-mediated cell elongation [35]. Brassinosteroids are well-known for their multifaceted roles in plant growth and development, as well as stress responses [70]. They are also known to affect shoot and root gravitropic responses [71–73]. This suggests a potential role for GLR-mediated $Ca^{2+}$ signaling in BR signaling or responses, which could have broader implications on plant growth and adaptation.

GLRs have been implicated in mediating stress responses, development, and gravitropism [18,19,74]. Some GLRs were robustly demonstrated to function in responses to leaf wounding and defense signaling, including roles for signaling via the defense-related hormones jasmonic acid and salicylic acid [29,33,36]. This raises the alternative possibility that the *D13-1* allele might affect reduced plant stature via constitutive defense signaling. Prior work on the gene expression consequences in the *D13-1*/+ mutants did not detect increased accumulation of transcripts encoded by defense response genes in the mutant [17]. Future work elucidating the mechanism of growth control in *D13-1* will help us understand which pathways respond to the GLR signaling produced by *D13-1*/+ and affect the reduced stature of these mutants.

### Cornichon homolog*, tod1,* regulates phenotypic severity of *D13-1* mutants

*D13-1*/+ is a semi-dominant allele of a glutamate receptor whose phenotypic severity is highly sensitive to the genetic background [17]. We carried out a GWAS using F1 crosses of the mutant to a diversity panel, a FOAM population, to identify the genetic modifiers responsible for variation in this mutant's phenotypic severity. GWAS in the FOAM population identified natural variation at a locus, *tropotriskaideca1 (tod1),* that affected the severity of the *D13-1*/+ mutant phenotype. This locus encoded a homolog of the drosophila GLR-interacting protein CORNICHON [61,62]. *Cis*-regulatory expression variation in the accumulation of TOD1 transcripts correlated with phenotypic variance, establishing that *tod1* plays a regulatory role in the variation in mutant traits. Interaction between plant CORNICHON homologs and GLRs has been previously demonstrated in Arabidopsis where CNIH proteins are required to localize Arabidopsis GLRs to the plasma membrane or endomembranes [61]. Co-expression of At*GLR3.3* with At*CNIH* genes, At*CNIH1* and At*CNIH4*, in COS-7 cell lines demonstrated greater conductance of currents as compared to At*GLR3.3* expressed by itself, suggesting that CNIH proteins enhance the channel activity of GLRs [61]. The interaction between GLRs and CNIH is shared across multiple GLRs and ligands, as AtGLR3.4 also requires CNIHs to conduct currents in response to the application of Glu or Asn [22]. We propose that modulation of *tod1* expression impacts GLR ion channel function and contributes to the observed phenotypic variation in *D13-1* mutants in different maize backgrounds. If true, additional GLR mutants of maize should also be modified by *tod1* alleles.

CORNICHON proteins function as cargo receptors, facilitating the trafficking of various membrane proteins to their target membranes. The first cornichon protein was characterized in Drosophila for the transport of the transforming growth factor α (TGFα) from the ER to the oocyte surface during *Drosophila* oogenesis [62]. In mammals, cornichon proteins interact with the AMPA subtype of ionotropic receptors [63]. The role of cornichons in the trafficking of multiple cation

transporters is conserved in plants. In rice, cornichon *OsCNIH1* interacts with sodium transported OsHKT1;3 [75]. In mosses, CNIHs are involved in the polar localization of auxin transporter PINA to the plasma membrane [76]. CORNICHON proteins not only promote trafficking of GLRs but also act as auxiliary subunits of these receptors modulating the kinetics of these receptors. In animals, binding of CORNICHON proteins to GLRs depends on the receptor's subunit composition. CNIH proteins in mice selectively bind to GluA1 subunits in GluA1A2 heteromeric AMPA-subtype of glutamate receptors, promoting surface expression of GluA1-containing AMPA receptors [63]. It is unclear whether such specificity also exists in plants. If the effects of CNIH are common across GLRs, we expect that *tod1* should suppress other dominant GLR mutants irrespective of the biological process in which the GLR is involved. If the effects are limited to *D13-1*, or a subset of GLRs, *tod1* alleles should not act as modifiers of other GLR mutants.

## GLR transcript accumulation patterns identify mechanisms of *D13-1/+* phenotypic expression variation

The *d13* gene encodes a GLR ion channel that is homologous to mammalian ionotropic glutamate receptors [17]. Like their mammalian counterparts, plant GLRs assemble as tetramers to form functional ion channels that facilitate $Ca^{2+}$ transport across the cell membrane [18,19,22]. The precise subunit composition of plant GLRs remains largely unknown. Studies in Arabidopsis indicate the existence of both homo- and hetero-tetrameric assemblies. Homomer formation for AtGLR1.1 and AtGLR3.4 has been observed when expressed in mammalian HEK293 cells [77]. The resolved structure of AtGLR3.4 confirmed its homo-tetrameric nature [22]. Despite this, single epidermal and mesophyll cells obtained from Arabidopsis leaves showed co-expression of five to six AtGLRs, suggesting the potential for hetero-tetramer formation [78]. Experimental approaches like yeast two-hybrid screens and Förster-resonance energy transfer (FRET) analysis in HEK293 cells, demonstrated physical interactions between various GLRs, further supporting the likelihood of heteromerization [77]. It is unknown whether D13 forms a homo- or hetero-tetramer. Suppression of the *D13-1/+* phenotype could result from dilution of the mutant subunit in a tetrameric channel by the wild-type protein encoded by wild-type *d13* allele in the heterozygote or variation at GLRs.

We employed a candidate gene analysis approach to test the effects of natural variation at the *d13* gene and other GLRs on *D13-1/+* phenotype expression. Natural variants at the *d13* locus affected mutant phenotypes and *d13* expression. In all cases, greater accumulation of the wild-type D13 transcript suppressed *D13-1/+* mutant phenotypes, consistent with *D13-1* encoding an aberrant subunit that can be diluted out of a multimeric GLR complex by greater concentrations of wild-type gene product. Linkage analysis of the SNPs and their effects at this locus revealed at least four *cis*-regulatory alleles affecting D13 transcript accumulation, not all of which affected mutant phenotypes. Either the recombination events decouple the effects of certain alleles on transcript accumulation from impacts on phenotypic traits, or the gene expression in certain cell types or stages affects phenotypic variation. Whatever the ultimate molecular mechanism of this variation, the presence of multiple segregating variants revealed complex genetic interactions and allelic diversity at this locus in maize.

The *d13* gene is strongly co-expressed with *glr7*. Phylogenetic analysis identified *d13* and *glr7* as syntelogs, arising from the recent maize-specific whole genome duplication [17], further supporting a close evolutionary and potentially functional relationship. Small effects on mutant height traits were observed at *glr7* at a relaxed threshold of p-value ≤ 1x10$^{-4}$. *Cis*-acting natural variation affecting the transcript abundance of GLR7 or other maize GLRs did not affect the *D13-1* phenotype, even at the relaxed p-value ≤ 1x10$^{-4}$. The phenotypic impact of alleles at *glr7* and evolutionary analyses suggest that GLR7 could be a partner for D13*.* Natural variation at the other GLRs did not affect the mutant traits specifically but did have weak effects on wild type height traits or affected both mutant and wild type heights similarly.

## Structural hypothesis for the *D13-1* effect on channel function

The mechanism by which *D13-1* induces dwarfism is unknown in both a structural biology and signaling pathway sense. Our genetic experiments, prior work on mutants of GLRs in other systems, and the available crystal and cryo-EM structures of GLRs indicate likely effects of the *D13-1* allele on channel function. *D13-1* encodes an allele that mutates an

alanine residue highly conserved in all GLRs in every biological system from prokaryotes to eukaryotes [17]. A mutation at this conserved position within the channel gating motif would alter the $Ca^{2+}$ pore formed by the GLR tetramer. Mutations in this same channel gating motif, though not this exact change, are well characterized in other systems [41–43]. Among the most well studied of these mutants is the semi-dominant *Lurcher* allele of mice, which results in a constitutively open channel disrupting $Ca^{2+}$ homeostasis and neurological function [41]. It is possible that *D13-1* also encodes a hyperactive channel. If so, suppression of the *D13-1* mutation by greater accumulation of TOD1 is expected to stabilize the GLR in the closed conformation when TOD1 is bound to a GLR tetramer, rather than through increased trafficking of the mutant sub-unit. Future work on the structural biophysics of the protein encoded by the *D13-1* allele, protein-protein interactions, and the operations of plant GLRs will deepen our understanding of these molecules, which we demonstrate, for the first time, can encode dominant alleles that affect plant growth.

### Dramatic suppression of *D13-1/+* by genetic background and phenotypic instability

A complete suppression of *D13-1/+* mutant phenotypes, where the mutants were indistinguishable from their congenic wild-type siblings, was observed in some F1 families within the FOAM population. Similar suppression of *D13-1/+* was previously observed in *D13-1/+* mutants in the Mo17 genetic background [17]. In Mo17, suppression was studied across multiple generations, and *D13-1* homozygotes were indistinguishable from wild-type plants in the Mo17 background [17]. The mechanism behind this complete suppression of *D13-1* mutants in Mo17 is unclear but suppression of homozygotes demonstrates a trans-acting effect. While natural variation affecting *tod1* expression modulated the phenotypic severity of the *D13-1* phenotype in the FOAM population, this effect was modest and is unlikely to explain the complete suppression observed in Mo17 [17] and multiple *D13-1/+* F1 families in our FOAM population. Additional genetic modifiers, complex epistatic interactions, or instability of expression of the *D13-1* allele itself may contribute to phenotypic suppression in some inbred backgrounds.

Because plant height was variable within F1 families, we measured the height traits of up to five mutant plants. Mean values for mutant heights in these families diluted the influence of extreme phenotypes, perhaps providing a better esti-mate of the modification by the background. If the spontaneous suppression of mutant phenotypes is due to instability at the *D13-1* allele, inclusion of suppressed family members might reduce our power to detect modifiers. To test this, we also carried out our GWAS using the height of the shortest mutant within each F1 family. There was a strong correlation between SFHT and mutant FIHT ($r^2 = 0.99$), and SFHT performed marginally better than mutant FIHT at the *d13* locus in our GWAS experiments suggesting that variability of mutant expression within the F1 families contributed to the error term in the association tests. This is consistent with stochastic epigenetic regulation of *d13* expression. Future work on the genome-wide consequences of epigenetic regulation of gene expression and testing of *D13-1* phenotypic expression in epigenetic mutants will permit testing of this hypothesis.

### Consistent modeling of SNP effects in GWAS is crucial for interpreting the effects of variants on multiple traits

GWAS serves as a powerful tool to identify natural genetic variants associated with specific traits. We can infer the potential relationship between these genetic variants and observed phenotypes by examining the directionality of SNP effects. When integrated with eGWAS, this allows us to determine the impact of those genetic variants on gene expression. A shared association strongly hints at a regulatory role, where the SNP modulates gene expression that, in turn, impacts the phenotypic trait. But the direction of the effect of each association is critical to constructing and testing hypotheses about molecular mechanism. For accurate cross-comparison of the results from independent GWAS experiments, it is imperative to evaluate the effects of the same allele on the traits of interest. While it seems trivial to state that mechanistic hypotheses, for example about gene expression and mutant phenotypes, require knowing which alleles increase or decrease transcript abundance and what effects that has on a visible phenotype, software used

for GWAS does not always consistently orient the effect direction estimates in a way that permits multiple experiment comparison. Moreover, molecular genetics tests of gene function benefit from anchoring effect directions relative to known genotypes for biochemists and molecular biologists. Some GWAS packages code the effect direction according to the major/minor allele frequencies. Different subsamples of the same population can result in higher-frequency allele switching between experiments, leading to incorrect assessments. Some GWAS packages code the effect direction by alphabetical order of the SNP genotype, which at least is consistent across populations and subsamples. We chose to orient our SNP effects relative to the B73 reference genome so that later mechanistic tests and work outside of population genetics were enabled. Establishing allele orientation using reference genome annotations ensures consistency and enhances reproducibility. We generated pre-numericalized SNP files, where the categorical data of SNP genotypes are represented as numerical values for reference and alternative SNP genotypes. Such a treatment is possible upstream of any GWAS method and would allow the comparison of SNP effects across experiments and populations regardless of the researcher's preferred GWAS method. [71,72]. By integrating the results from our phenotypic GWAS for *D13-1* traits with eGWAS for candidate genes, we were able to establish a functional relationship between the alleles and trait variation.

### Implications of trait correlation on GWAS results and genetic insights

Plant height variation in both mutant and wild type F1 plants were strongly correlated within the families in the FOAM population. The F1 families with shorter wild-type plants were likely to have more severe *D13-1/+* mutant phenotypes. This indicated that many genes, and micro-environmental plot-level variation, height had similar consequences in both wild type and mutant plants. These associations are not epistatic modifiers of the mutant and are not assignable to the pathway affected by *D13-1*. To establish epistatic relationships between natural variants and *D13-1*, and thereby place natural alleles within that pathway, the families were treated as case-control comparisons. We calculated difference and ratio traits for each F1 family to help highlight mutant-specific effects. We simultaneously carried out associations with mutant and wild-type traits and identified associations that affected mutant traits and not wild-type traits. Together, this identified variation at *d13* and *tod1* as affecting the difference, ratio, and mutant traits but not wild-type traits, supporting our assessment of these as inter-allelic and epistatic interactions, respectively.

### Conclusion

This study demonstrates that plant GLR and their interactors control growth in plants. The *D13-1* allele is encoded by a novel mutation in the gating domain of a GLR [17]. This domain forms a hinge in the protein, encoded by SYTAANLA in the *lurcher* gene of mouse [41] and SYTASLTS in *d13*. Mutation of the conserved alanine in the maize GLR produced a semi-dominant allele, similar to effects seen in animal alleles encoded by mutations in this domain including *lurcher* in mouse [41]. Constructing an experimental F1 association mapping population by crossing *D13-1/+* to an association panel detected alleles that modified the phenotype of the *D13-1* mutants but did not affect trait variation in wild-type siblings. These alleles' phenotypic effects were epistatically controlled by *D13-1/+*. Modifiers of the *D13-1/+* phenotype were encoded by cis-acting alleles in the *d13* gene itself and the *tod1* locus. The *tod1* locus encoded a maize homolog of the CORNICHON protein, a known interactor of animal and plant GLRs [61,63]. Unlike the 108 SNP-to-phenotype associations in the NHGRI-EBI GWAS Catalog (accessed 12/15/2025; [79]) at the four CORNICHON homologs of humans, which could be affected by any of the trafficking roles that these proteins play, the *tod1* association impacts a phenotype driven by a GLR. A linkage analysis of molecular and phenotypic data identified multiple alleles affecting both expression of the wild-type *d13* gene and modification of the *D13-1/+* phenotype. By comparing SNP linkage data, expression data, and phenotypic modification we were able to test proposed mechanisms for suppression of the mutant phenotype across independently inherited alleles demonstrating that increased accumulation of the wild-type D13 transcript could partially suppress the *D13-1/+* phenotype.

## Materials and methods

### Plant material

The maize *D13-1* allele came from an M1 population of EMS mutagenized B73 [17]. Due to the instability of the mutant phenotype in the B73 genetic background, only severe and intermediate *D13-1/+* mutants were used for crossing. Pollen from severe and intermediate *D13-1/+*:B73 mutants was crossed to 224 maize diverse lines to generate an $F_1$ association-tion mapping population (FOAM) during the summer of 2019. $F_1$ progeny were planted in two replications in a completely randomized design in the summer of 2020. Each $F_1$ progeny obtained from these crosses segregated 1:1 for wild-type and mutant siblings. Inbred lines B73, Mo17, CML322 and progenies of B73 x *D13-1/+*:B73, CML322 x *D13-1/+*:B73, and *D13/+*:B73/Mo17 x Mo17 were included as controls. Plants were grown at the Purdue Agronomy Center for Research and Education (ACRE) in West Lafayette, Indiana. Each plot was 3.84 meters (m) long and contained 16 kernels. The length of the alley and inter-row spacing were 0.79 m each. Standard fertilization, weed suppression, and pest control practices for maize cultivation were followed.

### Phenotypic data

For each F1 family in the FOAM population, three to five mutant plants and three wild-type plants were selected at random for phenotyping. A score of 0–5 was assigned to each $F_1$ family six weeks after planting based on the severity of the mutant phenotype. The $F_1$ families with severe mutants were assigned a score of 5, and those with mutants indistinguishable from wild type were assigned a score of 0.

Plant height, referred to as flag leaf height (FlHT), was measured as distance (in cm) from the soil line to the base of the topmost leaf/flag leaf at maturity. The ear height (EaHT) of mature plants was measured as the distance (in cm) from the soil line to the primary ear node. For F1 families where the mutant phenotype was completely suppressed, making the mutant and wild-type siblings indistinguishable, height measurements were taken from the most wild-type set of individuals, and a randomly selected set of individuals assigned as mutant. Ratios of mutant to wild-type heights and the differences between wild-type and mutant heights were also determined for height traits in each F1 family. In addition, flag leaf height for the shortest mutant (SFHT) in each F1 family was recorded. All the trait measurements for F1 families were collected in two replications and then averaged to calculate the mean trait values. The average measurements for all traits are provided in S1 Table.

### Statistical analysis and graphs

Frequency distributions of phenotypic traits in the FOAM population were tested for normality using the Anderson-Darling test [80] calculated using 'nortest' package in R [81]. Histograms for frequency distribution were plotted using the 'ggplot2' package in R [82]. Pairwise correlation among mutant and wild-type traits was calculated using the Pearson correlation coefficient. The correlation matrix was plotted using the 'chart.Correlation' function from the 'PerformanceAnalytics' package in R [83].

### Genome wide association

SNP data for the maize association panel were obtained from the imputation of the maize HapMap 3.2.1 data [84] as previously described [49]. This fully imputed SNP dataset contained 55,242,281 SNP positions across 343 inbred lines. To streamline computation and simplify the interpretation of allelic effects, SNPs were recoded with reference B73 alleles coded as an "A" and the alternate allele as a "T", and then numericalized [49]. This results in all allelic effects being expressed in the direction of the effect of the non-reference allele. SNPs with a minor allele frequency (MAF) <0.05 for each population were excluded to reduce false positives. Phenotypic data were collected for 224 F1 families, and the minor allele filtering for this set of individuals resulted in 22400498 SNPs being used for GWAS. For eGWAS, expression data was available from 296 maize lines, and minor allele filtering resulted in 23,913,217 SNPs being used in the analysis.

Expression data comprising Box-Cox transformed expression counts of D13 transcript and other GLRs from four tissues were obtained from a previously published RNA-sequencing study [64]. The four tissues included germinating shoot (GShoot), base of the third collared leaf (L3Base), mature leaf collected during day (LMAD), and mature leaf collected during night (LMAN). Genome-wide associations were performed by modifying the approach taken in switchgrass [85] to adapt the *bigsnpr* package [86] for maize. SNP associations at a threshold of p-value ≤ 1x10$^{-4}$ were retained in supplemental files to permit downstream hypothesis testing and data re-use with minimal false negative rates. Bonferroni corrected significance thresholds [87] were calculated based on the total number of tests for phenotypic GWAS (p-value = 0.05/22400498 = 2.2x10$^{-9}$) and eGWAS (p-value = 0.05/23913217 = 2.1x10$^{-9}$). SNPs located within 50 kb of the start or stop codon of the gene were considered *cis*-acting, and those located at 1Mb or higher from the gene were considered *trans*-acting. GWAS and locus plots were generated using the R package 'ggplot2' [82].

Potential candidate genes were tabulated for all GWAS SNPs associated with *D13-1* phenotypic traits by selecting up to three genes upstream and downstream from a 125 kb window around each SNP position (S5 Table). Each gene was ranked based on the distance of its start site to the SNP, with 1 being the closest and 3 being the farthest. Gene annotations were obtained from Gramene [88] and MaizeGDB (https://www.maizegdb.org) [89,90].

### Linkage disequilibrium and SNP clustering analysis

Estimates of LD for all pairwise SNP combinations were calculated using the 'LD' function in the 'gaston' package in R [91]. The results were visualized using the 'LD.plot' function in the 'gaston' package in R. Hierarchical clustering analysis was performed for SNPs within the *d13* locus associated with *D13-1* phenotypes and/or D13 transcript abundance, using the LD estimates ($r^2$). Pairwise genetic distances were computed using the Euclidean method, and SNP clustering was performed with the 'hclust' function in R version 4.2.2, employing the 'ward.D2' method [92]. The dendrogram was visualized using 'ggdendro' [93] and 'ggplot2' packages in R.

### Admixture-based GWAS

Accessions from the extant members of the two ancestral populations that gave rise to modern maize, *Zea mays* ssp. *parviglumis* and *Zea mays* ssp. *mexicana*, were used as source populations to estimate the admixture within our maize diversity set. These included randomly selected 30 unadmixed accessions from each of the two source populations described previously [94]. Genotypic data for these accessions for 31718584 SNPs were obtained from [95]. Of these SNPs, data for 6598906 SNPs were available for our maize diversity set. After filtering the SNPs for MAF < 0.05, the remaining 2741140 SNPs were used for ancestry estimation. Ancestry estimates for our set of diverse maize lines were inferred for each chromosome using ELAI [96] with the following parameters: two upper clusters, ten lower clusters, 20 EM steps, and 6000 generations. SNPs with missing rates larger than 0.1 were also removed. Admixture estimates were then input into GAPIT [97] as genotypic data and used to perform admixture state GWAS for *D13-1* phenotypic traits.

### Supporting information

**S1 Fig. Natural variation at *tod1* affects *D13-1* height.** SNPs at *tod1* and neighboring loci on chromosome 6 associated with mutant height traits in *D13-1* FOAM population at p-value ≤ 1x10$^{-4}$. A) Difference of wild type and mutant flag leaf height (Diff FIHT); B) Ratio of mutant to wild type flag leaf height (Ratio FIHT); C) Mutant flag leaf height (MT FIHT); D) Difference of wild type and mutant ear height (Diff EaHT); E) Ratio of mutant to wild type ear height (Ratio EaHT); F) Mutant ear height (MT EaHT).Top (lowest p-value) SNPs associated with each trait within the *tod1* locus are labelled. The grey dashed line depicts the genome-wide Bonferroni corrected p-value threshold.
(TIF)

**S2 Fig. Multiple *cis*-regulatory alleles at *d13* locus affect D13 transcript abundance and mutant height.** A) SNPs within ±50 kb of *d13* gene associated with D13 transcript abundance in GShoot and/or difference SFHT (Diff SFHT) at p-value ≤ 1x10⁻⁴. Grey dots represent SNPs that were not significantly associated with either of the two traits at p-value ≤ 1x10⁻⁴. The top SNPs associated with both traits are labelled; B) Scatter plot depicting the relationship between SNP effects on D13 transcripts in GShoot and Diff SFHT. Dashed trend line indicates a negative correlation between SNP effects on two traits; C) LD Plot for SNPs within ±50 kb of *d13* gene affecting D13 transcript abundance in GShoot and/or Diff SFHT at p-value ≤ 1x10⁻⁴. Blue bars at the top represent gene models. SNPs are highlighted based on trait associations at p-value ≤ 1x10⁻⁴. Boxes below the SNPs depict the direction of effects on traits, where blue indicates a positive effect and yellow indicates a negative effect of SNP on the trait. Colored bars below these mark different haplotypes; D) Hierarchical clustering of SNPs based on LD estimates (r²). Colored boxes represent different haplotypes.
(TIF)

**S3 Fig. Multiple *cis*-regulatory alleles at *d13* locus affect the D13 transcript abundance in L3Base and mutant severity score.** A) SNPs within ±50 kb of *d13* gene associated with D13 transcript abundance in L3Base and/or Mutant severity score at p-value ≤ 1x10⁻⁴. Grey dots represent SNPs that were not significantly associated with either of the two traits at p-value ≤ 1x10⁻⁴. The top SNPs associated with both traits are labelled; B) Scatter plot depicting the relationship between SNP effects on D13 transcripts in L3Base and Mutant severity score. Dashed trend line indicates a negative correlation between SNP effects on two traits; C) LD Plot for SNPs within ±50 kb of *d13* gene affecting D13 transcript abundance in L3Base and/or Mutant severity score at p-value ≤ 1x10⁻⁴. SNPs are highlighted based on trait associations at p-value ≤ 1x10⁻⁴. Boxes below the SNPs depict the direction of effects on traits, where blue indicates a positive effect and yellow indicates a negative effect of SNP on the trait. Colored bars below these mark different haplotypes; D) Hierarchical clustering of SNPs based on LD estimates (r²). Colored boxes represent different haplotypes.
(TIF)

**S4 Fig. Co-expression of *d13* and *glr7* in maize.** Scatter plot represents the relative expression of *d13* and *glr* in maize tissues accessed from the ATTED II co-expression database.
(TIF)

**S5 Fig. *Cis*-regulatory variation affecting transcript accumulation of Maize GLRs.** SNPs within ±50kb of each Maize GLR associated with variation in transcript accumulation at p-value ≤ 1e-4.
(TIF)

**S1 Table. Phenotypic data for *D13-1*/+ FOAM population.**
(XLSX)

**S2 Table. SNPs at *tod1* locus associated with mutant severity score and plant height traits at p-value ≤ 1x10⁻⁴.**
(XLSX)

**S3 Table. All SNPs within ±250 kb of *cnih1* and their effects on phenotypic traits in *D13-1* FOAM population.**
(XLSX)

**S4 Table. SNPs within the *tod1* locus associated with mutant phenotypic traits and TOD1 transcript abundance in four above-ground tissues at p-value ≤ 1x10⁻⁴.**
(XLSX)

**S5 Table. SNPs associated with mutant severity score and plant height traits in *D13-1* FOAM population at a p-value ≤ 1x10⁻⁴.**
(XLSX)

**S6 Table. Candidate genes for SNPs associated with phenotypic traits *D13-1* FOAM population at a p-value ≤ 1x10$^{-4}$.**
(XLSX)

**S7 Table. SNPs at *d13* locus (±250kb) associated with variation in phenotype in *D13-1* FOAM population at a p-value ≤ 1x10$^{-4}$.**
(XLSX)

**S8 Table. *Cis* SNPs associated with D13 transcript abundance in four above-ground tissues at p-value ≤ 1x10$^{-4}$.**
(XLSX)

**S9 Table. All SNPs within ±50kb of *d13* gene and their effects on D13 transcript abundance in different tissues and/or phenotypic traits.**
(XLSX)

**S10 Table. List of Maize GLR genes.**
(XLSX)

**S11 Table. SNPs within ± 250kb of Maize GLRs affecting the phenotypic traits in *D13-1* FOAM population at p-value ≤ 1x10$^{-4}$.**
(XLSX)

**S12 Table. SNPs at Maize GLRs associated with D13 transcript abundance in trans at p-value ≤ 1x10$^{-4}$.**
(XLSX)

**S13 Table. *Cis* SNPs associated with transcript abundance of Maize GLR at p-value ≤ 1x10$^{-4}$ in four above-ground tissues.**
(XLSX)

**S14 Table. Admixture GWAS associations with *D13-1* phenotypic traits at p-value ≤ 1x10$^{-4}$.**
(XLSX)

## Acknowledgments

We gratefully acknowledge the leadership (Jim Beaty, Rachel Stevens, and Jason Adams) and staff at Agronomy Center for Research and Education (ACRE) at Purdue University, West Lafayette, IN, for their assistance with planting and managing the field experiments described in this study. We thank the Rosen Center for Advanced Computing (RCAC) for providing essential computing resources. We thank James Booker and Nino Ferrer for compositions and Lisa Kekaula and Bob Vennum for encouraging us to put ourselves in so we could get something out. We are grateful to Ed Buckler and his lab at USDA, ARS for generating the maize gene expression dataset and making it publicly accessible for research like ours. Special thanks to Jeffrey Ross-Ibarra for providing teosinte subspecies SNP data sets that permitted admixture calculations for the association panel, as well as insightful discussions on admixture mapping. We further acknowledge the National Science Foundation of the United States of America for supporting open-source data sharing through the CyVerse platform.

## Author contributions

**Conceptualization:** Amanpreet Kaur, Rajdeep S. Khangura, Brian P. Dilkes.

**Data curation:** Amanpreet Kaur, Rajdeep S. Khangura.

**Formal analysis:** Amanpreet Kaur, Rajdeep S. Khangura, Brian P. Dilkes.

**Funding acquisition:** Rajdeep S. Khangura, Brian P. Dilkes.

**Investigation:** Amanpreet Kaur, Rajdeep S. Khangura, Brian P. Dilkes.

**Methodology:** Amanpreet Kaur, Rajdeep S. Khangura, Brian P. Dilkes.

**Project administration:** Brian P. Dilkes.

**Resources:** Brian P. Dilkes.

**Supervision:** Brian P. Dilkes.

**Visualization:** Amanpreet Kaur, Rajdeep S. Khangura.

**Writing – original draft:** Amanpreet Kaur, Brian P. Dilkes.

**Writing – review & editing:** Amanpreet Kaur, Rajdeep S. Khangura, Brian P. Dilkes.

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
