## [Decision Letter · Decision Letter 0]

28 Oct 2025

PGENETICS-D-25-00922

The semi-dominant D13-1 mutant of maize is modified by strong cis-acting natural variation in transcript abundance.

PLOS Genetics

Dear Dr. Dilkes,

Thank you for submitting your manuscript to PLOS Genetics. After careful consideration, we feel that it has merit but does not fully meet PLOS Genetics's publication criteria as it currently stands. Therefore, we invite you to submit a revised version of the manuscript that addresses the points raised during the review process.

Please submit your revised manuscript within 60 days Dec 27 2025 11:59PM. If you will need more time than this to complete your revisions, please reply to this message or contact the journal office at plosgenetics@plos.org. Please include the following items when submitting your revised manuscript:

We look forward to receiving your revised manuscript.

Kind regards,

Tatiana Giraud

Academic Editor

PLOS Genetics

Quan Wang

Section Editor

PLOS Genetics

Aimée Dudley

Editor-in-Chief

PLOS Genetics

Anne Goriely

Editor-in-Chief

PLOS Genetics

**Additional Editor Comments:**

We have now received two referee reports for your manuscript on cis-acting natural variation in transcript abundance impacting the phenotype of the semi-dominant D13-1 mutant of maize, which encodes a defective ionotropic glutamate receptor. Both referees found the manuscript valuable and original; however, concerns were raised, in particular about the clarity of the objectives and main results of the study, and the lack of broader context in the discussion.

We agree that the study would be a nice addition to PLoS Genetics but that the goals and novelty of the paper are unclear. It would be better to begin the abstract and the introduction by general statements on the scientific topics rather than on specificities of the mutants, avoiding abbreviations, especially without defining them, to provide clearer take-home messages on the results and their importance, and to render globally the abstract and title easier to understand for non-specialist readers. The introduction and discussion also could better convey the main results and their importance.

I would encourage resubmission if you are able to revise the manuscript along these lines. The referees also provide a list of excellent additional suggestions, which should also be addressed.

**Journal Requirements:**

At this stage, the following Authors/Authors require contributions: Amanpreet Kaur, Rajdeep Singh Khangura, and Brian P Dilkes. Please ensure that the full contributions of each author are acknowledged in the "Add/Edit/Remove Authors" section of our submission form.

The list of CRediT author contributions may be found here: https://journals.plos.org/plosgenetics/s/authorship#loc-author-contributions

https://journals.plos.org/plosgenetics/s/submission-guidelines#loc-parts-of-a-submission

4) We are unable to open the following Supporting Information file: Supplemental Figures 1-5.zip. Please kindly revise as necessary and re-upload.

Potential Copyright Issues:

i) Please confirm (a) that you are the photographer of 2A, or (b) provide written permission from the photographer to publish the photo(s) under our CC BY 4.0 license.

ii) Figure 1B. Please confirm whether you drew the images / clip-art within the figure panels by hand. If you did not draw the images, please provide (a) a link to the source of the images or icons and their license / terms of use; or (b) written permission from the copyright holder to publish the images or icons under our CC BY 4.0 license. Alternatively, you may replace the images with open source alternatives. See these open source resources you may use to replace images / clip-art:

7)  Please send a completed 'Competing Interests' statement, including any COIs declared by your co-authors. If you have no competing interests to declare, please state "The authors have declared that no competing interests exist". Otherwise please declare all competing interests beginning with the statement "I have read the journal's policy and the authors of this manuscript have the following competing interests"

**Reviewers' comments:**

Reviewer's Responses to Questions

**Comments to the Authors:**

Reviewer #1: The paper entitled “The semi-dominant D13-1 mutant of maize is modified by strong cis-acting natural variation in transcript abundance”, provide evidence strengthening the link between glutamate receptor-like channels pathway and growth regulation in plants. And by using a population genetics approach, the authors deepen our understanding on how genetic variation can regulate a complex trait.

In one hand, the paper yield information about GLR11/D13-1 allele regulation in maize. Such as the modulation of GLR11 transcript abundance due to particular SNPs located in cis. Or by the very plausible interaction with a CORNICHON stabilizing protein (encoded by the Zm00001d036084 gene), which constitutes proof of GLRs and CNIH interaction for the first time in maize (at least at gene level). Using as a model the GLRs pathway, in the other hand, some of the effects of the genetic variation over a polygenic and complex trait are revealed.

However, the main objective of the authors is unclear. The title, the abstract and the paper organization, seem to me inclined to the second (using the GLRs pathway as a model to study the influence of natural variation in a selected trait). But the introduction and the discussion fail delimiting and organizing both subjects so that the reader can follow the main theme. Alternatively, if the aim is focused on the GLR11 gene characterization, the introduction must deepen in GLRs prior characterization and functions (specially their involvement with distinct hormones, since the link with growth may reside there).

Although some of the results reported on the paper have already been observed in other plant species as Arabidopsis, Physcomitrium and rice (as the interaction of CNIH or their role in balancing defense and growth), it does contain novel data for Zea mays. Which contributes to the characterization and classification of this important receptors. I consider that the strongest aspects of the paper are related to GLR characterization in maize (including the cis-acting regulation revealed by natural genetic variation). And I highlight that the authors offer a list of the GLR genes in maize, a basic foundation for further studies in the area. The study of genetic variation influence on a trait, is also interesting. However, it remains uncertain whether a general model of the mechanisms linking genetic variation and gene activity can be established based on these findings. Since it appears to be highly diverse and depending on the characteristics of the specific gene.

Major issues.

1. The core theme of the manuscript needs to be more clearly articulated. Both the Introduction and Discussion sections should be revised to consistently align with and reinforce this central theme.

2. The manuscript would benefit significantly from a clearer delineation of its overall direction and the specific subjects explored. While the interconnected nature of these aspects is important, clearly setting these boundaries will make the paper more accessible to readers and effectively highlight the impact of the claims presented.

3. Despite aiming to establish the influence of genetic variation on a selected trait as a main topic, the introduction and neglect to establish the frame to contextualize the subject significance. Although the it does talk briefly about the link between genetic variation and biological pathway regulation, several other prior topics as GLR functions or plant growth regulation dilutes the attention from this central theme. I suggest to always place first the central theme, and once is fully presented or discussed, particularities of the secondary subject can be addressed in a deep but concise way.

4. The discussion fails to offer broad conclusions on mechanisms of how variation in DNA sequences can influence a trait. Instead it alternates the focus on the particularities of GLR regulation (such as the formation of tetramers or the known characteristics and functions of CORNICHON genes). To avoid diluting the relevance of how genetic variation influencing traits, GLR specific results can be used as examples to make general points.

Alternatively, the main focus of the paper can be GLR11 gene regulation and function in maize. While the linkage of natural genetic variation and trait regulation can come as secondary theme. In this case, GLR11 phenotype can be broadly discussed in the context of known GLR functions.

5. Following this idea, the paper could benefit from a deeper integration with evidence of GLR activities in other species. Although the paper from Yi, et al. (2022) is mentioned, there are other papers linking GLRs with defense and (directly or indirectly) with growth. For example, the papers showing that the systemic signaling against herbivory is mediated by the glutamate receptor-like channels (Mousavi, et al. 2013), demonstrate the link with the Jasmonate pathway. Strong pieces of evidence accumulated over the years (such as growth defects in jasmonate signaling mutants), have sustained the defense-growth trade-off. So that the involvement of GLRs upstream jasmonate signaling and in defense in general (Li, et al. 2013 & Toyota, et al. 2018), indirectly link them with growth. The work from Hernández-Coronado, et al (2022) represents another piece of evidence linking the GLRs and growth. By showing that regeneration is mediated by GLRs through the SA pathway, the relationship of GLRS with distinct hormone pathways beyond Brassinosteroids may account for the growth defect phenotype of the D13-1 mutant.

Recommended references.

Yu, B., Wu, Q., Li, X., Zeng, R., Min, Q. and Huang, J. (2022), GLUTAMATE RECEPTOR-like gene OsGLR3.4 is required for plant growth and systemic wound signaling in rice (Oryza sativa). New Phytol, 233: 1238-1256.

Mousavi SA, Chauvin A, Pascaud F, Kellenberger S, Farmer EE. GLUTAMATE RECEPTOR-LIKE genes mediate leaf-to-leaf wound signalling. Nature. 2013 Aug 22;500(7463):422-6. doi: 10.1038/nature12478. PMID: 23969459.

Li F, Wang J, Ma C, Zhao Y, Wang Y, Hasi A, Qi Z. Glutamate receptor-like channel3.3 is involved in mediating glutathione-triggered cytosolic calcium transients, transcriptional changes, and innate immunity responses in Arabidopsis. Plant Physiol. 2013 Jul;162(3):1497-509. doi: 10.1104/pp.113.217208.

Hernández-Coronado M, Dias Araujo PC, Ip PL, Nunes CO, Rahni R, Wudick MM, Lizzio MA, Feijó JA, Birnbaum KD. Plant glutamate receptors mediate a bet-hedging strategy between regeneration and defense. Dev Cell. 2022 Feb 28;57(4):451-465.e6. doi: 10.1016/j.devcel.2022.01.013.

6. The name of both the GLR gene and the CORNICHON gene, bust be maintained as the names previously attributed in the literature. So that the D13-1 mutant should be named as ZmGLR11-1. And the TOD1 gene as ZmCNIH. I do understand that those mutants were found by forward genetics and named according to the experiment. But once the identity if corroborated, names can be changed for clarity and uniformity in nomenclature, as well as to give credit to previous research. Finally, the impact of the paper can be increased by having the GLR name in the title.

Minor issues:

1. Introduction for GLRs cites several reviews, but not original papers. As mentioned before, I think the paper can benefit from a deeper discussion of GLR functions. For which original research papers characterizing GLR functions in plants may come useful.

2. The introduction could be more concise in its presentation of growth as a complex trait. While this point is important, the explanation is longer than necessary and could be tightened to improve flow and reader engagement.

3. The introduction can be improved by addressing further the role of genetic variation in gene expression and regulation by relevant studies in yeast and humans.

4. Page 9, last sentence. Zm00001d03608 is missing a number.

“Transcript abundance data for the Zm00001d03608 gene in the inbreds constituting the maize association panel were obtained from four above-ground tissues [germinating shoot…”

Reviewer #2: In this paper, the authors are investigating the molecular bases of the semi-dominant dwarf maize mutant D13-1, caused by a SNP in the coding sequence that mutate an otherwise conserved alanine into a guanine in the hinge that opens and closes the gate of a glutamate receptor-like ion channel (GLR). As this phenotype is semi-dominant and can be modified or even completely reversed in some genetic backgrounds, it is most likely a complex phenotype partially determined by some complex regulatory interactions. The authors thus used a F1 association mapping approach by crossing a D13-1 mutant with hundreds of maize lines to investigate the molecular bases of dwarfism associated with D13-1. They performed association studies with dwarfism and gene expression levels to uncover modifier loci, i.e. the mutations playing a regulatory role and affecting the dwarf phenotype. They identified one trans-modifier loci, tod1, and several independent cis-regulatory loci on the d13 locus that affect the mutant severity. The study is sound and well-presented, I only have minor comments.

Comments

1. Please reference the subpanel of the figure when citing it in the text, it will make it easier for the reader to navigate the paper (especially for figures 4-6 and supplementary figure S1, where there are a lot of similar panels). Please also elaborate on the content of the different panels (in particular what the trats are) in Supp. Fig.1.

2. For the single locus association study at d13: why did you choose a 250kb window and not a larger one, as cis-regulatory relationships can span up to 1Mb in maize?

3. For the eQTL study at d13 locus: why announcing an eGWAS approach and then referencing eQTLs in the text. You can directly say that you used an eQTL discovery approach, that would be less confusing for the reader.

4. p.9 last paragraph, “Transcript abundance data for the Zm00001d03608 gene in the inbreds”: there is a “4” missing at the end of the gene name.

5. In Figure 4, the E. is missing in the legend.

6. Please put all occurrence of cis and trans in italic.

7. Please add a data availability section with links or references to all genomic and raw expression data if they are stored in public databases. If pertinent, please mention the date at which the data were dowloaded and genome version. For the scripts, please mention the version of the packages used. Making the script publicly available would enhance the repoducibility of the paper.

**Have all data underlying the figures and results presented in the manuscript been provided?**

Reviewer #1: Yes

Reviewer #2: **No: ** I think all the data used are publicly available, but there are no "Data availability" section that allows to easily find them.

PLOS authors have the option to publish the peer review history of their article (what does this mean? ). If published, this will include your full peer review and any attached files.

**Do you want your identity to be public for this peer review?** For information about this choice, including consent withdrawal, please see our Privacy Policy .

Reviewer #1: No

Reviewer #2: No

**Figure resubmission:**
---

## [Decision Letter · Decision Letter 1]

11 Dec 2025

PGENETICS-D-25-00922R1

A maize mutant in the glutamate receptor-like dwarf13 is modified by cis-acting natural variation and a Cornichon homolog

PLOS Genetics

Dear Dr. Dilkes,

Thank you for submitting your manuscript to PLOS Genetics. After careful consideration, we feel that it has merit but does not fully meet PLOS Genetics's publication criteria as it currently stands. Therefore, we invite you to submit a revised version of the manuscript that addresses the points raised during the review process.

Please submit your revised manuscript within by Jan 10 2026 11:59PM. If you will need more time than this to complete your revisions, please reply to this message or contact the journal office at plosgenetics@plos.org. Please include the following items when submitting your revised manuscript:

We look forward to receiving your revised manuscript.

Kind regards,

Tatiana Giraud

Academic Editor

PLOS Genetics

Quan Wang

Section Editor

PLOS Genetics

Aimée Dudley

Editor-in-Chief

PLOS Genetics

Anne Goriely

Editor-in-Chief

PLOS Genetics

**Additional Editor Comments:**

I was pleased to see that this manuscript has been carefully revised, being much clearer, and the referees also found the revisions overall satisfactory, they only had a few minor additional suggestions left. I also have additional suggestions below.

-Lines numbers are missing, which renders cumbersome to make comments for editors and referees

-The wording can be wording, some examples are given below, but the text needs to be carfully edited.

-“semi-dominant mutants”: it is not the mutant or the locus that is dominant, but an allele, right? Correct all along the manuscript

-Abstract: explain why this mutant is interesting, what is the phenotype earlier in the abstract

-P1: “segregated for mutant and wild-type F1 hybrids”: alleles segregate, not hybrids (same P4); “epistatically controlled” does not seem right (sounds like the opposite? The loci control the expression?); “linked” is ambiguous: is it physical linkage or more generally within the pathway or similar? “encoded by natural variation” is incorrect, a locus is not encoded and natural variation in itself does not encode anything; integration of different results not of integration.

-P2: the severity applies to the phenotype not to the mutant; what does semidominant mean? Codominant? Unclear what you mean by “provides insights into mechanisms shaped by natural selection”? what mechanisms? (same in P4). The author summary is much clearer and better written.

-P5: “the allele segregates as a single locus” sounds like a pleonasm, and an allele is not a locus.

-P6: “genetic suppression” does not seem a correct formulation; of what?

-P7: again, suppression of what? “suppressed lines” also sounds incorrect, it is not the lines that are suppressed but the expression of a gene?

-P10: a gene does not encode another gene

-P11: the severity does not apply to heterozygotes but to their phenotype

-Keep homogeneous tenses in sentences (for example P11 we assessed… if variation WAS linked”, but also elsewhere)

-P13: recombination occurs between loci not between phenotypes (here transcript level and height)

-P17: “were likely contributed” does not seem correct syntax

-Do not refer to figures o tables in the discussion, they should have been explained well enough in the result section

-P19: the consequence on gene regulation of the mutation not the mutant

-The discussion remains hard to read for non-specialists..; It would be good to add a first paragraph and/or a conclusion broadening the context and explaining the importance of the results for the broad audience of PloS Genetics

**Journal Requirements:**

**Reviewers' comments:**

Reviewer's Responses to Questions

**Comments to the Authors:**

Reviewer #1: In this version of the paper, now entitled "A maize mutant in the glutamate receptor-like dwarf13 is modified by cis-acting natural variation and a Cornichon homolog”, the authors succeed in delivering a clear and concise story that contributes to the advance of maize genetics. The two aspects included in the original paper are still present: the regulation of a complex developmental trait by natural genetic variation; and the characterization of members of the GLR family (particularly dwarf13 and its regulation by tod1). But in this version, the ideas are well organized, making easier to follow and to understand the paper. The introduction, for example, suffered a restructuration, making it shorter and concise. While the conclusion now highlights the relevance of the findings.

I am satisfied with the answers of the authors to my concerns. They reply to each one in a very logic and simple way, and took into consideration most comments or suggestions. I really hope that this review process has contributed to strengthen the story.

I only have a minor comment:

On page 16, second paragraph, there is an error. It says, “in in all four tissues” (in is repeated).

Reviewer #2: The authors have answered all of my questions, and have updated the abstract, introduction and discussion in a way that makes it both more understandable and more attractive for the reader.

**Have all data underlying the figures and results presented in the manuscript been provided?**

Reviewer #1: Yes

Reviewer #2: Yes

PLOS authors have the option to publish the peer review history of their article (what does this mean? ). If published, this will include your full peer review and any attached files.

**Do you want your identity to be public for this peer review?** For information about this choice, including consent withdrawal, please see our Privacy Policy .

Reviewer #1: No

Reviewer #2: No

**Figure resubmission:**
---

## [Editor Report · Decision Letter 2]

20 Dec 2025

Dear Dr Dilkes,

We are pleased to inform you that your manuscript entitled "A maize mutant in the glutamate receptor-like dwarf13 is modified by cis-acting natural variation and a Cornichon homolog" has been editorially accepted for publication in PLOS Genetics. Congratulations!

Yours sincerely,

Tatiana Giraud

Academic Editor

PLOS Genetics

Quan Wang

Section Editor

PLOS Genetics

Aimée Dudley

Editor-in-Chief

PLOS Genetics

Anne Goriely

Editor-in-Chief

PLOS Genetics

BlueSky: @plos.bsky.social

Comments from the reviewers (if applicable):

**Data Deposition**

http://datadryad.org/submit?journalID=pgenetics&manu=PGENETICS-D-25-00922R2

**Press Queries**

---

## [Editor Report · Acceptance letter]

PGENETICS-D-25-00922R2

A maize mutant in the glutamate receptor-like dwarf13 is modified by cis-acting natural variation and a Cornichon homolog

Dear Dr Dilkes,

We are pleased to inform you that your manuscript entitled " 

A maize mutant in the glutamate receptor-like dwarf13 is modified by cis-acting natural variation and a Cornichon homolog" has been formally accepted for publication in PLOS Genetics! Your manuscript is now with our production department and you will be notified of the publication date in due course.

With kind regards,

Zsofia Freund

PLOS Genetics

On behalf of:
